# The cryptic gonadotropin-releasing hormone neuronal system of human basal ganglia

**Katalin Skrapits[1]\*, Miklós Sárvári[1], Imre Farkas[1], Balázs Göcz[1], Szabolcs Takács[1], Éva Rumpler[1], Viktória Váczi[1], Csaba Vastagh[2], Gergely Rácz[3], András Matolcsy[3], Norbert Solymosi[4], Szilárd Póliska[5], Blanka Tóth[6], Ferenc Erdélyi[7], Gábor Szabó[7], Michael D Culler[8], Cecile Allet[9], Ludovica Cotellessa[9], Vincent Prévot[9], Paolo Giacobini[9], Erik Hrabovszky[1]\***

[1]Laboratory of Reproductive Neurobiology, Institute of Experimental Medicine, Budapest, Hungary; [2]Laboratory of Endocrine Neurobiology, Institute of Experimental Medicine, Budapest, Hungary; [3]1st Department of Pathology and Experimental Cancer Research, Semmelweis University, Budapest, Hungary; [4]Centre for Bioinformatics, University of Veterinary Medicine, Budapest, Hungary; [5]Department of Biochemistry and Molecular Biology, Faculty of Medicine, University of Debrecen, Debrecen, Hungary; [6]Department of Inorganic and Analytical Chemistry, Budapest University of Technology and Economics, Budapest, Hungary; [7]Department of Gene Technology and Developmental Biology, Institute of Experimental Medicine, Budapest, Hungary; [8]Amolyt Pharma, Newton, France; [9]Univ. Lille, Inserm, CHU Lille, Laboratory of Development and Plasticity of the Neuroendocrine Brain, Lille Neuroscience & Cognition, Lille, France

**\*For correspondence:**
skrapits.katalin@koki.hu (KS);
hrabovszky.erik@koki.hu (EH)

**Competing interests:** The authors declare that no competing interests exist.

**Abstract** Human reproduction is controlled by ~2000 hypothalamic gonadotropin-releasing hormone (GnRH) neurons. Here, we report the discovery and characterization of additional ~150,000–200,000 GnRH-synthesizing cells in the human basal ganglia and basal forebrain. Nearly all extrahypothalamic GnRH neurons expressed the cholinergic marker enzyme choline acetyltransferase. Similarly, hypothalamic GnRH neurons were also cholinergic both in embryonic and adult human brains. Whole-transcriptome analysis of cholinergic interneurons and medium spiny projection neurons laser-microdissected from the human putamen showed selective expression of *GNRH1* and *GNRHR1* autoreceptors in the cholinergic cell population and uncovered the detailed transcriptome profile and molecular connectome of these two cell types. Higher-order non-reproductive functions regulated by GnRH under physiological conditions in the human basal ganglia and basal forebrain require clarification. The role and changes of GnRH/GnRHR1 signaling in neurodegenerative disorders affecting cholinergic neurocircuitries, including Parkinson's and Alzheimer's diseases, need to be explored.

## Introduction

Mammalian reproduction is controlled by a few hundred/thousand preoptic/hypothalamic neurons, which release the decapeptide neurohormone gonadotropin-releasing hormone (GnRH) into the hypophysial portal circulation. GnRH promotes fertility via increasing the synthesis and secretion of luteinizing hormone and follicle-stimulating hormone in the anterior pituitary (*Herbison, 2018*). Unlike other neurons of the central nervous system, GnRH neurons are born in the olfactory placodes and migrate into the forebrain prenatally (*Casoni et al., 2016*; *Schwanzel-Fukuda and*

*Pfaff, 1989*; *Wray et al., 1989*). Recent developmental studies on embryos/fetuses determined the detailed spatio-temporal profile of this process in the human (*Casoni et al., 2016*). Approximately 2000 neurons were observed along a ventral migratory path whereby GnRH neurons reach the hypothalamus to regulate reproduction after puberty. In addition, a previously unknown dorsal migratory route has been identified whereby ~8000 GnRH neurons migrated towards pallial and/or subpallial structures. The fate of these neurons at later stages of pre- and postnatal development has been unexplored so far.

While GnRH neurons in adult laboratory rodents are mostly preoptic/hypothalamic and serve reproductive functions (*Merchenthaler et al., 1980*), a handful of anatomical studies on primates identified additional *GNRH1* mRNA expression and/or GnRH immunoreactivity in extrahypothalamic regions unrelated to reproduction. These included several basal ganglia and the basal forebrain (*Krajewski et al., 2003*; *Quanbeck et al., 1997*; *Rance et al., 1994*; *Terasawa et al., 2001*). Initial enthusiasm to study these neurons further faded after suggestions that extrahypothalamic GnRH neurons in monkeys contain the GnRH degradation product GnRH1-5, instead of the *bona fide* GnRH decapeptide (*Quanbeck et al., 1997*; *Terasawa et al., 2001*).

Here, we study human extrahypothalamic GnRH neurons in adult *postmortem* brains with immunohistochemistry (IHC), in situ hybridization (ISH), single-cell transcriptomics (RNA-seq), and high-performance liquid chromatography/tandem mass spectrometry (HPLC-MS/MS). We report and characterize a previously unexplored large GnRH neuron population with ~150,000–200,000 cell bodies scattered in different basal ganglia and the basal forebrain. Extrahypothalamic GnRH neurons, most of which are found in the putamen (Pu), contain *bona fide* GnRH decapeptide, as shown by HPLC-MS/MS. Deep transcriptome analysis reveals that these neurons express GnRH biosynthetic enzymes, *GNRHR1* autoreceptors, and the molecular machinery of cholinergic and GABAergic cotransmission. Somewhat unexpectedly, hypothalamic GnRH neurons also exhibit cholinergic phenotype in the embryonic and adult human brains. Altogether, these data indicate that GnRH is synthesized as a co-transmitter of many cholinergic neurons in the human basal ganglia and basal forebrain. At least in the Pu, GnRH appears to act on GnRHR1 autoreceptors to regulate higher-order non-reproductive functions associated with the cholinergic system.

## Results

### Human extrahypothalamic GnRH-immunoreactive neurons occur in the basal ganglia and the basal forebrain

The primate central nervous system contains extrahypothalamic GnRH cell populations, which have unknown functions (*Krajewski et al., 2003*; *Quanbeck et al., 1997*; *Rance et al., 1994*; *Terasawa et al., 2001*). An earlier ISH study of adult human brains identified ~6000–7000 *GNRH1* mRNA-expressing neurons in the Pu and the nucleus basalis magnocellularis of Meynert (nbM), among other sites (*Rance et al., 1994*). Here, we used IHC to address the presence and map the distribution of GnRH-immunoreactive (IR) neurons in extrahypothalamic sites of three adult human brains (#1–3). Every 24th 100-µm-thick coronal section between Bregma levels −22.5 and 33.1 (*Mai et al., 1997*) was immunostained using a well-characterized guinea pig antiserum (#1018) against GnRH decapeptide (*Hrabovszky et al., 2011*; *Figure 1A, B*). This experiment revealed numerous extrahypothalamic GnRH-IR neurons in the Pu, moderate numbers in the nucleus accumbens (nAcc) and the head of the nucleus caudatus (Cd), and lower numbers also in the nbM (*Figure 1C*). Labeled neurons were also scattered in the globus pallidus (GP), the ventral pallidum (VP), and the bed nucleus of the stria terminalis (BnST). The immunostained perikarya showed round or oval shape, with a mean diameter of 29 µm in the Pu (*Figure 1B, C*). Preabsorption of the working solution of this antiserum with 0.1 µg/ml GnRH decapeptide eliminated all labeling in control experiments using sections of three subjects (#17–19) (*Figure 2A*).

### Quantitative analysis detects ~ 150,000–200,000 extrahypothalamic GnRH neurons in the adult human brain most of which are located in the putamen

GnRH neurons develop in the olfactory placodes and migrate to the brain prenatally (*Schwanzel-Fukuda and Pfaff, 1989*; *Wray, 2001*). Recent studies from Casoni and colleagues identified 10,000

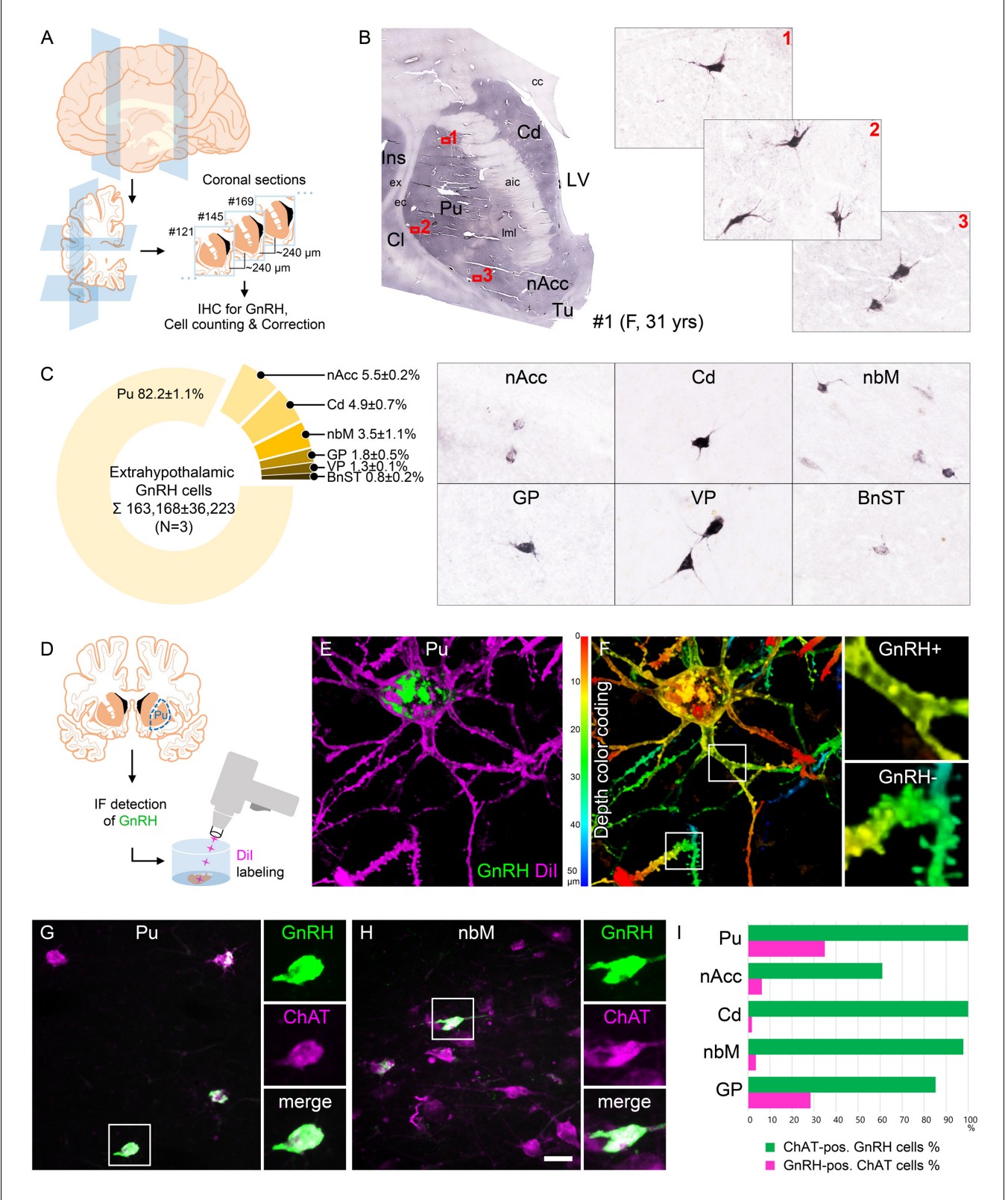

**Figure 1.** Anatomical approaches unveil the distribution, number, fine structure, and cholinergic phenotype of extrahypothalamic gonadotropin-releasing hormone (GnRH) neurons in the adult human brain. (**A**) Extrahypothalamic GnRH-immunoreactive (GnRH-IR) neurons were mapped with immunohistochemistry (IHC) and quantified in the brain of three adult human individuals (#1–3). (**B**) Representative coronal section of a 31-year-old female subject (#1) illustrates the caudate nucleus (Cd), putamen (Pu), claustrum (Cl), insular cortex (Ins), anterior limb of the internal capsule (aic),

*Figure 1 continued on next page*

*Figure 1 continued*

external capsule (ec), extreme capsule (ex), corpus callosum (cc), lateral medullary lamina (lml), lateral ventricle (LV), nucleus accumbens (nAcc), and olfactory tubercle (Tu). High-power insets (1–3) reveal extrahypothalamic GnRH neurons many of which can be found in the Pu. (C) The majority (82.2%) of the 163,168 ± 36,223 extrahypothalamic GnRH neurons occurred in the Pu, followed by the nAcc, Cd, nucleus basalis magnocellularis (nbM), globus pallidus (GP), ventral pallidum (VP), and bed nucleus of the stria terminalis (BnST). (D) To visualize the fine structure of dendrites, the immunofluorescent (IF) detection of GnRH was combined with cell membrane labeling using DiI delivered with a Gene Gun. (E) 3-D reconstruction of the DiI-labeled (magenta) GnRH-IR (green) neurons revealed large multipolar cells, which exhibited only few dendritic spines. (F) Depth color coding, where colors represent distance from the section surface, allowed better distinction between DiI-labeled processes of the GnRH neuron (upper inset; GnRH+) from other DiI-labeled neuronal elements many of which belonged to medium spiny GABAergic projection neurons (lower inset; GnRH-). (G) Double-IF experiments addressed the presence of the cholinergic marker enzyme choline acetyltransferase (ChAT) in GnRH neurons. Nearly all GnRH neurons in the Pu contained ChAT signal. (H) The GnRH neuron population also overlapped with cholinergic projection neurons of the nbM. (I) With few exceptions, GnRH neurons were ChAT-immunoreactive (green columns), whereas they represented relatively small subsets of cholinergic cells (magenta columns) being the highest in the Pu (~35%). Scale bar (shown in **H**): 3.5 mm in low-power photomicrograph (**B**), 50 μm in (**B, C, G and H**) (insets in **G** and **H**: 25 μm), 12.5 μm in (**E**) and (**F**) (insets: 3.7 μm).

The online version of this article includes the following figure supplement(s) for figure 1:

**Figure supplement 1.** Cell numbers determined with light microscopic analysis require compensation for overcounting using Abercrombie's correction factor.

migrating GnRH neurons in human embryos/fetuses most of which (~8000) followed a previously unknown dorsal migratory route targeting subpallial and/or pallial structures, as opposed to the ~2000 neurons in the ventral route leading to the hypothalamus (*Casoni et al., 2016*). We addressed the possibility that extrahypothalamic GnRH-IR neurons of the adult brain originate from the 8000 neurons observed in the dorsal pathway. With this aim, we determined the total number of GnRH-IR neurons in the basal ganglia and the basal forebrain. Immunolabeled neurons were counted in every 24th section of a single hemisphere using light microscopy (*Figure 1A, B*). Cell counts were then multiplied by 24 and 2 (for the two hemispheres) and compensated for overcounting (*Abercrombie, 1946*; *Guillery, 2002*; *Figure 1—figure supplement 1A*). The total number of extrahypothalamic GnRH neurons calculated this way in three subjects was 229,447 (31-year-old female; #1), 155,357 (61-year-old male; #2), and 104,699 (62-year-old male; #3) (163,168 ± 36,223; mean ± SEM). These unexpectedly high GnRH cell numbers made it unlikely that human extrahypothalamic GnRH neurons develop from olfactory placodes and migrate into the brain along the dorsal migratory route (*Casoni et al., 2016*). The individual variations in total GnRH cell numbers of the three samples may be both biological and technical, which would be difficult to separate. 82.2 ± 1.1% of labeled cells were observed in the Pu, 5.5 ± 0.2% in the nAcc, 4.9 ± 0.7% in the Cd, 3.5 ± 1.1% in the nbM, 1.8 ± 0.5% in the GP, 1.3 ± 0.1% in the VP, and 0.8 ± 0.2% in the BnST (*Figure 1C*).

## Extrahypothalamic GnRH neurons synthesize *bona fide* GnRH decapeptide derived from the *GNRH1* transcript

Results of previous studies with IHC on embryonic and fetal rhesus monkey brains questioned whether extrahypothalamic GnRH neurons synthesize *bona fide* GnRH decapeptide (*Quanbeck et al., 1997*; *Terasawa et al., 2001*). First, developing GnRH neurons in the septum, stria terminalis, amygdala, striatum, and internal capsule of the monkey brain were not detected by several GnRH antibodies (*Quanbeck et al., 1997*; *Terasawa et al., 2001*), including the widely used LR-1 rabbit GnRH polyclonal antiserum (*Silverman et al., 1990*). Second, these neurons exhibited immunoreactivity to EP24.15 (aka thimet oligopeptidase; THOP1), a metalloendopeptidase, which can cleave GnRH at the Tyr5-Gly6 position to generate GnRH1-5 (*Terasawa et al., 2001*). To investigate the possibility that GnRH neurons in the basal ganglia and the basal forebrain of the adult human brain use GnRH1-5, rather than GnRH decapeptide for signaling, we first tested a series of polyclonal antibodies against human GnRH-associated peptide (hGAP1) or GnRH decapeptide (*Supplementary file 2*) for their reactivity with GnRH neurons of the human Pu (N = 10; #5, 6, and 12–19). All of the tested antibodies, including the LR-1 antiserum, recognized GnRH-IR neurons (*Figure 2B*), suggesting that these cells contain the *bona fide* GnRH. Neurons detected with different antibodies were identical as they were double-labeled (*Figure 2C*) in dual-immunofluorescence (IF) experiments (N = 2; #4 and 5) using two GnRH antibodies from different host species. Results of further control experiments with the combined use of IF and non-isotopic ISH (N = 5; #15–19)

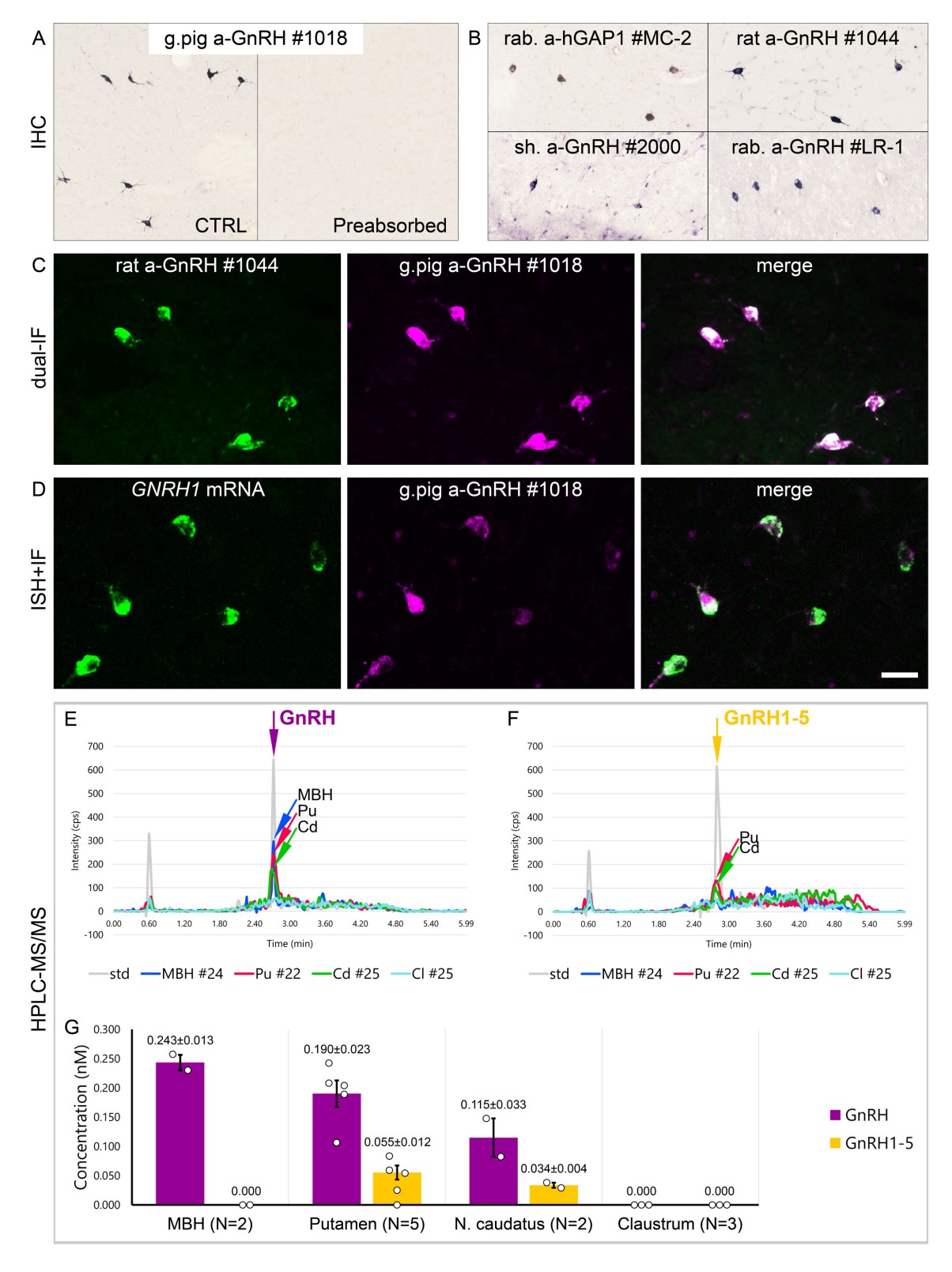

**Figure 2.** Combined evidence from immunohistochemistry (IHC), in situ hybridization (ISH), and high-performance liquid chromatography-tandem mass spectrometry (HPLC-MS/MS) indicates that extrahypothalamic gonadotropin-releasing hormone (GnRH) neurons synthesize *bona fide* GnRH decapeptide derived from the *GNRH1* transcript. (**A**) IHC labeling of the human putamen with the guinea pig polyclonal GnRH antiserum #1018 reveals a large number of immunoreactive neurons in control sections (Ctrl) of a 64-year-old male subject (#19). All labeling is eliminated if the working solution

*Figure 2 continued on next page*

*Figure 2 continued*

of #1018 is preabsorbed overnight with 0.1 µg/ml GnRH. (B) A series of additional primary antisera against the human GnRH-associated peptide (hGAP1) or GnRH also recognize immunoreactive neurons in the human putamen (#6). Such antibodies include the LR-1 rabbit primary antiserum, which was reported previously not to label extrahypothalamic GnRH neurons in the developing monkey brain. (C) Positive control with the combined use of two GnRH antibodies from different host species for dual-immunofluorescence (IF) experiments on sample #5 provides evidence that the antibodies detect the same neuronal elements. (D) Non-isotopic ISH/IF dual-labeling studies reveal that GnRH-immunoreactive neurons express *GNRH1* mRNA, indicating that extrahypothalamic GnRH is a *GNRH1* gene product (sample #15). (E) As illustrated in representative chromatograms, HPLC-MS/MS ( #21–28) detects *bona fide* GnRH decapeptide in tissue extracts from the mediobasal hypothalamus (MBH), putamen (Pu), and nucleus caudatus (Cd), but not the claustrum (Cl). (F) The GnRH1-5 degradation product is present in the Pu and Cd and undetectable in the MBH and Cl. (G) Quantitative analysis reveals the highest tissue concentrations of GnRH in the MBH, somewhat lower levels in the Pu and the Cd, and no detectable GnRH decapeptide signal in the Cl. Note that tissue concentrations of GnRH in the Pu and the Cd are 3–4 times higher than those of GnRH1-5. Scale bar (shown in D): 120 µm in (A, B), 50 µm in (C, D).

showed that GnRH-IR neurons express *GNRH1* mRNA (*Figure 2D*). Finally, to provide direct evidence for the biosynthesis of the GnRH decapeptide in these cells, tissue samples (#21–28) were microdissected from the mediobasal hypothalamus (MBH; N = 2), Pu (N = 5), Cd (N = 2), and Cl (N = 3). HPLC-MS/MS analysis of the tissue extracts established that the dominant peptide form in the Pu and Cd is the GnRH decapeptide. GnRH1-5 was also present, albeit at 3–4 times lower tissue concentrations (*Figure 2E–G*). Only GnRH decapeptide was detectable in the MBH (used as a positive control) where hypophysiotropic GnRH neurons occur and neither peptide form was present in the Cl, in accordance with the absence of IHC labeling at this site (*Figure 2E–G*). Together with observations from the IHC and ISH experiments, HPLC-MS/MS results gave firm support to the notion that extrahypothalamic GnRH neurons mainly produce *bona fide* GnRH decapeptide derived from the *GNRH1* gene.

## GnRH neurons of the putamen are large multipolar interneurons with smooth-surfaced dendrites

The IHC method was unable to visualize the entire dendritic arbor of extrahypothalamic GnRH-IR cells (*Figure 1B, C*). This limitation could be due to the low amount and/or restricted subcellular distribution of the peptide. To overcome this problem, we labeled the dendritic compartment of GnRH cells with the lipophilic dye DiI for further morphological analysis (*Takács et al., 2018*; *Figure 1D*). Following the immunofluorescent (IF) visualization of GnRH neurons in the Pu of a 72-year-old female (#20), DiI-coated tungsten particles were delivered into the sections using a Helios Gene Gun (Bio-Rad) (*Figure 1D*; *Takács et al., 2018*). Spreading of this lipophilic dye along the cytoplasmic membrane caused Golgi-like labeling of random-hit neurons, including 12 GnRH-IR cells (*Figure 1E, F*). Confocal microscopic analysis and 3-D reconstruction of the DiI signal revealed spider-like neurons with rich arborization of poorly spined dendrites. DiI-labeled GnRH neurons were clearly distinct from the main Pu cell type, the densely spined medium spiny GABAergic projection neurons (SPNs) (*Figure 1E, F*).

## Extrahypothalamic GnRH cells represent subpopulations of cholinergic neurons

SPNs represent 80–98% of striatal neurons, the remainder being made up of cholinergic and different subclasses of GABAergic interneurons (*Gonzales and Smith, 2015*). DiI-labeled GnRH cells resembled cholinergic interneurons (ChINs) in size and dendritic morphology. Indeed, dual-IF experiments (N = 4; #3–5 and 19) established that GnRH neurons of the Pu contain the cholinergic marker enzyme choline acetyltransferase (ChAT) (*Figure 1G*). Similarly, GnRH neurons in the nbM (*Figure 1H*) and other extrahypothalamic sites also exhibited ChAT immunoreactivity. The extent of ChAT/GnRH colocalization was assessed quantitatively in five distinct regions of a 62-year-old male subject (#3). Confocal microscopic analysis of representative dual-labeled sections established that the vast majority of extrahypothalamic GnRH neurons are cholinergic (green bars in *Figure 1I*). In contrast, GnRH-IR neurons represented only 34.9% of all cholinergic neurons in the Pu, 6.3% in the nAcc, 1.8% in the head of the Cd, 3.6% in the nbM, and 28.4% in the GP (magenta bars in *Figure 1I*). GnRH-positive and GnRH-negative cholinergic neurons often intermingled, without gross morphological differences between the two phenotypes (*Figure 1G, H*).

## Hypothalamic GnRH neurons regulating reproduction also exhibit an unexpected cholinergic phenotype

The ChAT phenotype emerged as a hallmark of extrahypothalamic GnRH neurons. To confirm this notion by verifying the absence of ChAT in the hypothalamic GnRH neuron population, hypothalamic tissue sections were processed for dual-IF detection of ChAT and GnRH, followed by confocal microscopic analysis. Unexpectedly, 34.6 ± 7.1% of the hypothalamic GnRH neurons also exhibited ChAT signal in seven adult human male and female subjects (*Figure 3A, B*; #3 and 6–11), a phenomenon not observed in other species before.

## Cholinergic phenotype of human GnRH neurons develops prenatally

To address when GnRH neurons gain the cholinergic phenotype, prenatal co-expression of ChAT and GnRH was explored via dual-IF experiments in coronal sections of two fetal heads (#29 and 30) at gestational week 11 (GW11). At this age, ~20% of GnRH neurons can still be found in the nasal region, whereas the majority have already entered the brain to migrate toward hypothalamic and extrahypothalamic target areas (*Casoni et al., 2016*). While GnRH-positive neurons within the nasal compartment did not contain ChAT signal (*Figure 3C*), those in the septum (*Figure 3D*), the striatum (*Figure 3E*) and elsewhere in the developing brain were already ChAT-IR. These data suggest that migrating GnRH neurons become cholinergic after entering the brain and continue to express ChAT immunoreactivity in hypothalamic as well as extrahypothalamic regions.

## Neurons laser-capture microdissected from the *postmortem* putamen provide sources for high-quality RNA suitable for RNA-seq

ChINs, at least one-third of which synthesize GnRH in the human Pu (*Figure 1I*), communicate with SPNs locally (*Ahmed et al., 2019*). To localize the receptors mediating the effects of GnRH in the adult human Pu, transcriptome profiling of cellular samples enriched in ChINs and SPNs was carried out. Being the largest cell type, ChINs were readily recognizable in sections subjected to Nissl-staining under RNase-free conditions, whereas the medium-sized SPNs represent the most frequently encountered neuronal phenotype in the Pu (*Figure 4A*). Laser-capture microdissection (LCM) was used to collect neuronal pools enriched in ChINs and SPNs from cresyl violet-stained Pu sections of two human subjects (#21 and 22). Each ChIN-enriched pool contained ~300 large neurons and each SPN-enriched pool consisted of ~600 medium-sized neurons (*Figure 4A*). Total RNA was isolated and RNA-seq libraries were prepared from the four cell pools and sequenced with the Illumina Next-Seq 500/550 High Output (v2.5) kit. 29.4M and 25.8M reads were obtained from the two ChIN pools and 26.3M and 3.2M reads from the two SPN pools. Approximately 9.6M and 6.6M reads from ChIN and ~4.3M and 0.35M reads from SPN pools were mapped to the GRCh38.p13 human reference genome; 13,664 and 12,637 identified transcripts occurred at cpm >5 in ChIN pools and 13,558 and 13,682 transcripts in SPN pools (*Figure 4A* and *Supplementary file 3*).

## Size-based laser-capture microdissection allows adequate sampling of striatal cholinergic interneurons and medium spiny projection neurons

Cholinergic markers, including *CHAT*, *SLC5A7*, *SLC18A3*, *ACHE,* and *CHRM2*, were highly enriched in the ChIN pools from subjects #21 and 22. These transcripts were either absent or found at low levels only in the two SPN pools (*Figure 4B*). Mouse ChINs arise from Nkx2.1+ progenitors. During development, Nkx2.1 upregulates the expression of the LIM homeobox proteins LHX8, ISL1, and GBX2, which, in turn, promote cell differentiation into ChINs (*Allaway and Machold, 2017*). These LIM transcripts as well as type 3 vesicular glutamate transporter (*SLC17A8*) showed robust and exclusive expression in ChINs (*Figure 4B*). The SPN pools, in turn, expressed much higher levels of known SPN markers than ChINs, including various cholinergic (*CHRM1*), serotonergic (*HTR6*), glutamatergic (*GRM1*), and dopaminergic (*DRD1*) receptor isoforms and several neuropeptides (*TAC1*, *PDYN*, *PENK*) (*Figure 4B*). Differential distribution of the above transcripts verified that the size-based LCM strategy efficiently separated ChINs from SPNs for transcriptome profiling. Relatively high levels of expression of known GABAergic marker transcripts (*GAD1*, *GAD2*, *SLC6A1,* and *SLC32A1*) in ChINs, in addition to SPNs (*Figure 4B*), revealed that ChINs use GABAergic co-transmission, as proposed earlier for ChINs of the rodent CPU (*Lozovaya et al., 2018*).

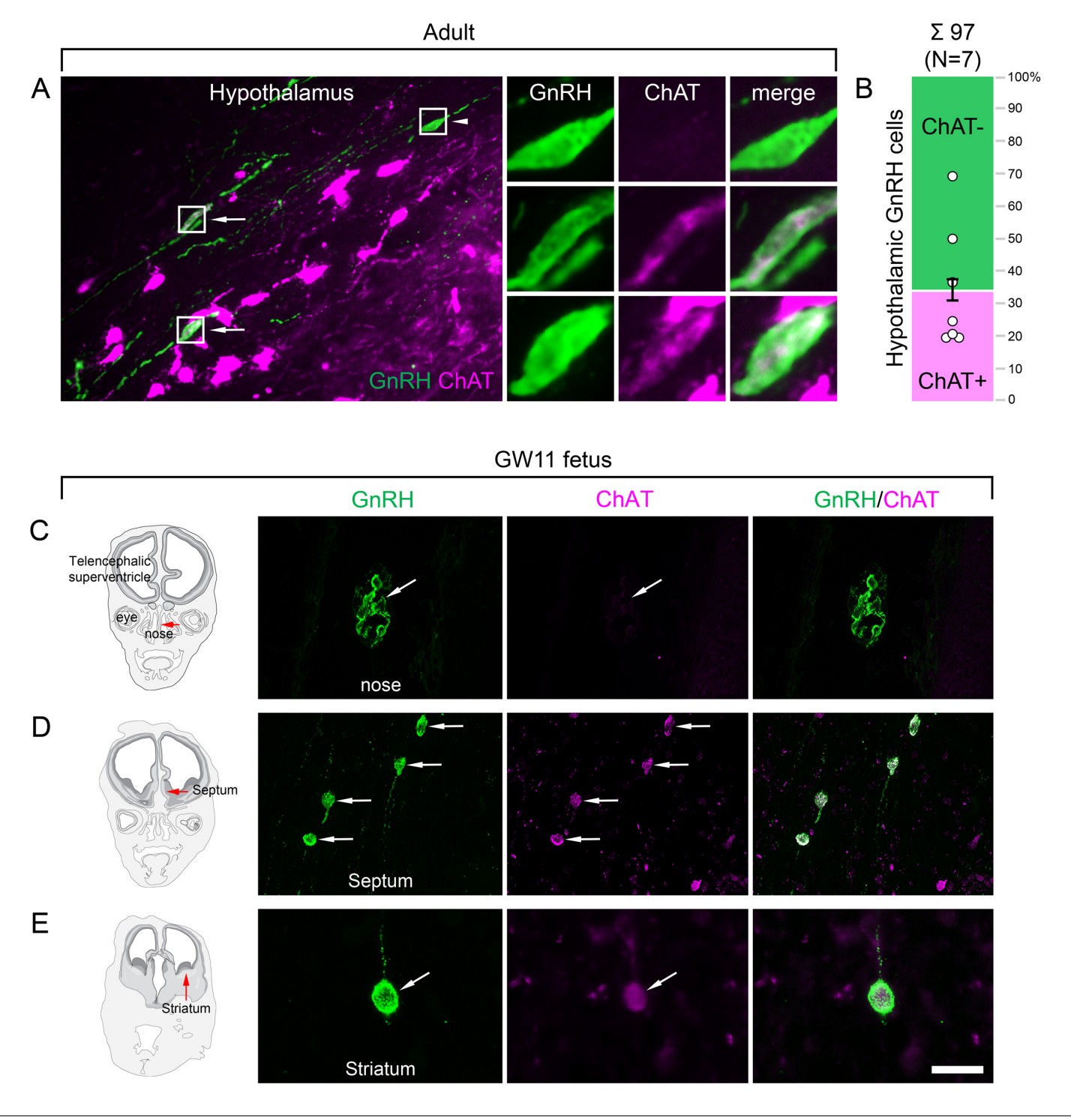

**Figure 3.** Both hypothalamic and extrahypothalamic gonadotropin-releasing hormone (GnRH) neurons exhibit cholinergic phenotype gained during early fetal development. (**A**) The cholinergic phenotype is not a hallmark of human extrahypothalamic GnRH neurons because large subsets of GnRH neurons in the adult human hypothalamus (green immunofluorescent signal; subject #3) also exhibit choline acetyltransferase (ChAT; magenta) immunoreactivity. High-power insets show single- (arrowhead) and dual-labeled (arrows) GnRH neurons from framed regions. (**B**) Quantitative analysis of 97 GnRH neurons from seven subjects (#3; 6–11) reveals the ChAT phenotype in 34.6 ± 7.1% of hypothalamic GnRH neurons. (**C–E**) The cholinergic phenotype of GnRH neurons is gained during early fetal development. Left panels illustrate coronal views of the fetal head at GW11 (#29). Representative photomicrographs taken from sites indicated by the red arrows show results of dual-immunofluorescence experiments. (**C**) At this stage of development, a large subset of GnRH neurons (green immunofluorescent signal) migrate in the nasal region toward the brain and do not exhibit

*Figure 3 continued on next page*

*Figure 3 continued*

ChAT signal. (D, E) In contrast, GnRH neurons migrating through the septal area (D, arrows) or located in the striatum (E, arrow) express ChAT (magenta). Scale bar (shown in E): 50 μm in (A, C, D) (insets in A: 12.5 μm), 20 μm in (E).

## Cholinergic interneurons selectively express *GNRH1* and *GNRHR1* and contain GnRH biosynthetic enzymes

*GNRH1* was expressed exclusively in the two ChIN pools, in accordance with the morphological observations (*Figure 4C*). Processing of the proGnRH1 protein begins with endoproteolysis by pro-hormone convertases from which ChINs abundantly expressed the *PCSK2* isoform. Enzymes catalyzing subsequent steps of GnRH biosynthesis, including carboxypeptidases (*CPE*, *CPD*), peptidylglycine α-amidating monooxygenase (*PAM*), and glutaminyl cyclase enzymes (*QPCT*), were also present in ChINs (*Figure 4C*). The promiscuous THOP1 enzyme accounts for the cleavage of multiple neuropeptides, including GnRH. This enzyme was expressed in both ChINs and SPNs, with a higher relative abundance in the latter. The most important finding of the RNA-seq study was that the seven-transmembrane receptor *GNRHR1* was expressed selectively in ChINs. This observation indicated that GnRH in the human Pu acts on GnRHR1 autoreceptors and also made it unlikely that SPNs are affected directly by GnRH derived from ChINs. Altogether, transcriptome profiling of ChINs and SPNs provided molecular support to a concept that GnRH is synthesized by ChINs and acts locally via GnRHR1 autoreceptors.

## Transcriptome profiling provides novel insight into the molecular connectome of the human putamen

Transcriptome profiling of ChINs and SPNs revealed a large set of genes that were expressed selectively or predominantly in one cell type only, in addition to many other genes expressed in both. Neurotransmitter and neurotransmitter receptor transcripts identified this way allowed us to propose signaling mechanisms that act in the bidirectional communication between ChINs and SPNs. Some receptors appear to serve as autoreceptors (e.g., *GNRHR1*, *NMBR*, *CRHR1/2*). Others may receive ligands from multiple neuronal sources within (e.g., *QRFPR*, *NPY1R/5R*, *TACR1*, *SSTR2/3*) or outside (e.g., *OXTR*, *MC4R*, *GLP1R*, *PRLR*) the striatum. Peptidergic mechanisms concluded from the transcriptome profiles are illustrated as a schematic model in *Figure 5*. A deeper insight into the molecular connectome of the human Pu can be obtained from the detailed receptor and neuropeptide expression profiles of ChINs and SPNs and from the full list of expressed genes in the two SPN and two ChIN cell pools (*Supplementary file 3*; BioProject accession number: PRJNA680536).

## Discussion

### Extrahypothalamic GnRH-IR neurons correspond to type III GnRH neurons detected earlier with in situ hybridization and synthesize the full-length GnRH decapeptide derived from the *GNRH1* gene

A pioneer ISH study by Rance and co-workers distinguished three types of *GNRH1* mRNA-expressing neurons in the human brain based on size, shape, and labeling intensity (*Rance et al., 1994*). The GnRH-IR neurons we detected in our study correspond to type III neurons characterized by round/oval shape, large nucleus and nucleolus, prominent Nissl substance, and *GNRH1* mRNA levels intermediate between those of heavily labeled type I neurons in the MBH and lightly labeled type II neurons in the medial septum and the dorsal medial preoptic area (*Rance et al., 1994*). Our IHC studies also detected many type I hypothalamic GnRH neurons but found only few septal type II neurons, which latter had negligible contribution to the total GnRH cell numbers.

Although type III GnRH neurons have not been reported in adult laboratory rodents, they occur in the striatum, amygdala, and nbM of non-human primates (*Krajewski et al., 2003*). These neurons were proposed to differ from the hypothalamic GnRH cell population in that they contain the GnRH1-5 degradation product of GnRH instead of the *bona fide* GnRH decapeptide (*Quanbeck et al., 1997*; *Terasawa et al., 2001*). Circumstantial IHC evidence to support this notion stemmed from IHC observations made on developing monkey embryos and fetal brains (*Quanbeck et al., 1997*; *Terasawa et al., 2001*). First, these neurons could not be immunolabeled

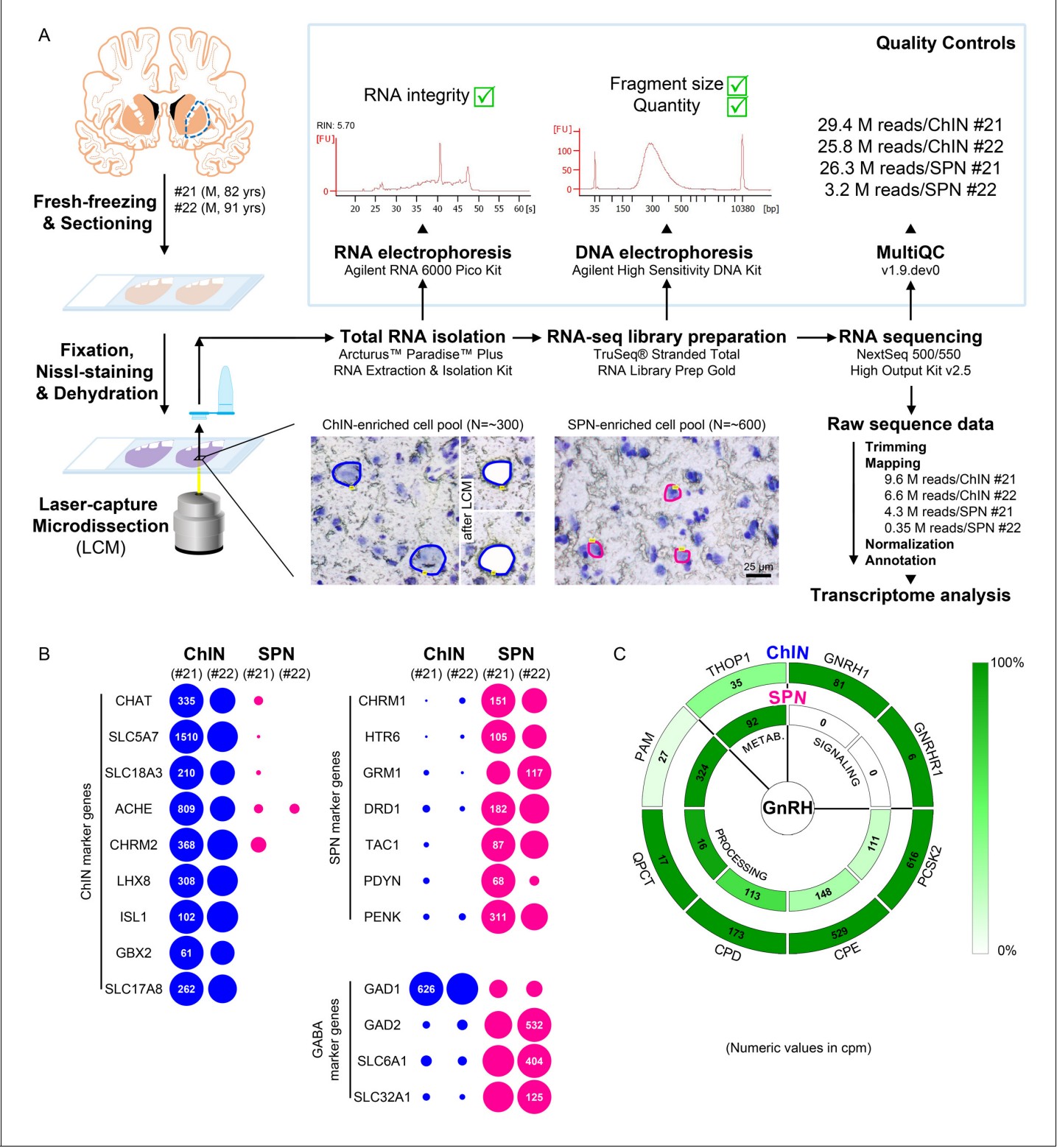

**Figure 4.** Deep transcriptome profiling of cholinergic interneurons (ChINs) and spiny projection neurons (SPNs) provides new insight into extrahypothalamic gonadotropin-releasing hormone (GnRH) signaling mechanisms and the molecular connectome of the human putamen. (A) 20-μm-thick coronal sections were collected on PEN membrane slides from frozen putamen samples of two male human subjects (#21 and 22) and fixed with an ethanol/paraformaldehyde mixture. Neurons were visualized using Nissl-staining and isolated with laser-capture microdissection (LCM). 300 neurons included in each ChIN-enriched cell pool were recognized based on their large perikaryon size. The vast majority of the 600 medium-sized

*Figure 4 continued on next page*

*Figure 4 continued*

microdissected and pooled neurons corresponded to SPNs, the major putamen cell type. Total RNA was isolated and RNA-seq library prepared from both cell types and sequenced with the Illumina NextSeq 500/550 High Output (v2.5) kit. (B) Bioinformatic analysis verified high enrichment of known cholinergic markers in the two ChIN pools and of SPN markers in the two SPN pools. Expression levels in dots reflect counts per million reads (cpm), and in each case, dot areas reflect transcript abundances relative to the highest cpm (100%). (C) Key elements of proGnRH processing, GnRH signaling, and GnRH metabolism are illustrated in two concentric circles. The *GNRH1* and *GNRHR1* transcripts are present in ChINs only (outer circle). ChINs express all enzymes required for proGnRH processing. The promiscuous *THOP1* enzyme, which may account for GnRH cleavage, occurs in both cell types, at higher levels in SPNs than in ChINs. Color coding reflects relative transcript abundances, whereas numbers indicate cpms (mean cpms of subjects #21 and 22).

with the LR-1 rabbit polyclonal antiserum and a series of other antibodies against GnRH (*Quanbeck et al., 1997*). Second, they showed weaker labeling with GnRH4-10 and heavier labeling with GnRH1-5 antibodies than migrating GnRH neurons targeting the hypothalamus (*Terasawa et al., 2001*). Third, they showed immunoreactivity to the THOP1 enzyme, which can cleave GnRH at the Tyr5-Gly6 position, although notably, they did not differ from hypothalamic GnRH neurons in this latter respect (*Terasawa et al., 2001*).

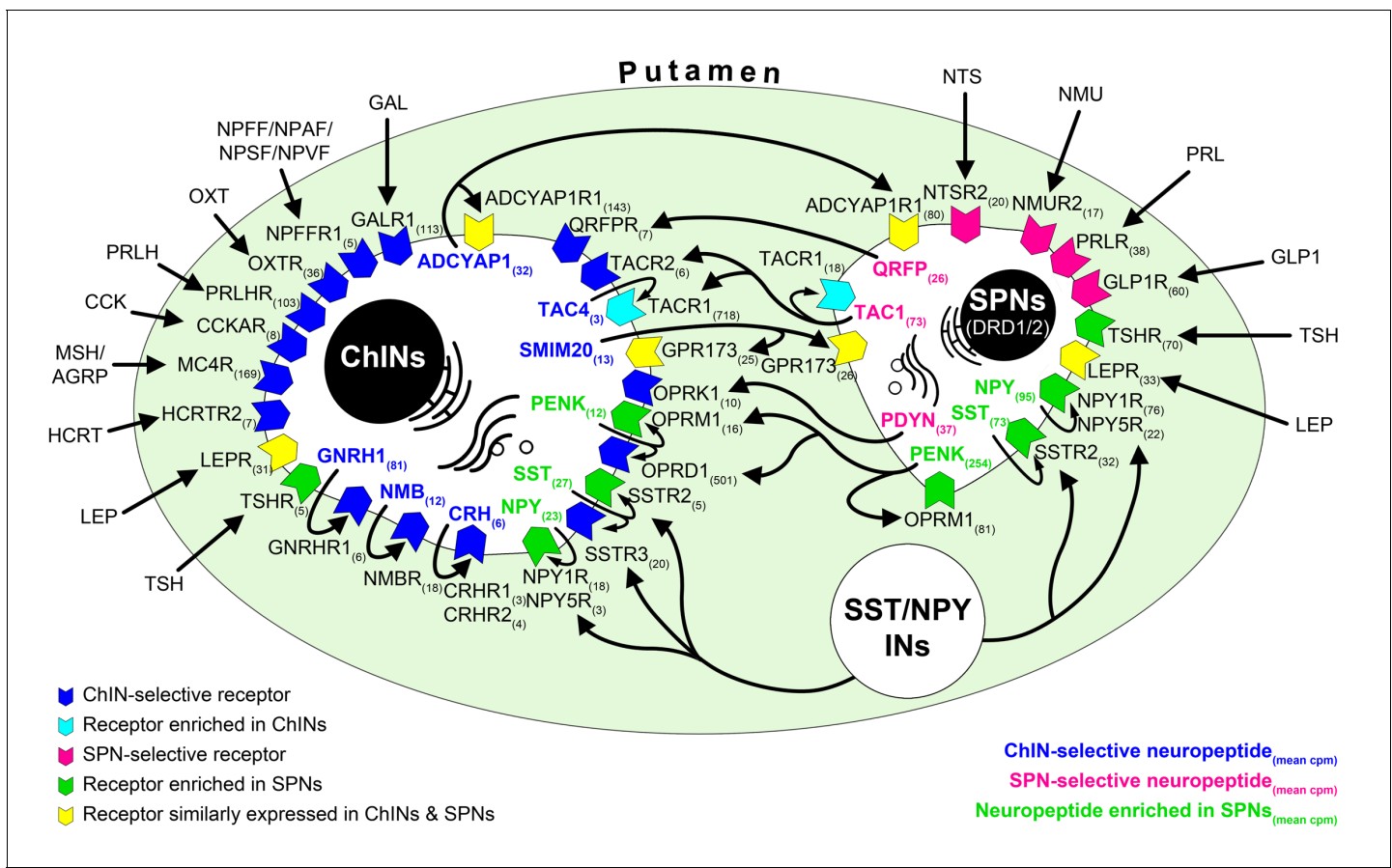

**Figure 5.** RNA-seq studies reveal the neuropeptide and neuropeptide receptor expression profiles of cholinergic interneurons (ChINs) and spiny projection neurons (SPNs) and provide insight into the molecular connectome of putamen cell types. Proposed signaling mechanisms are based on neuropeptide and peptide receptor expression profiles of the two cell types. ChINs appear to use *GNRHR1*, *CRHR1/2*, and *NMBR* autoreceptor signaling. *SSTR2*, *NPY1R/5R*, *OPRM1*, and *TACR1* may serve, at least partly, as autoreceptors in SPNs. Proposed peptidergic communication between the two cell types is also indicated by arrows. Other receptors receive ligands from different neuronal sources within (e.g., *QRFPR*, *NPY1R/5R*, *TACR1*, *SSTR2/3*) or outside (e.g., *OXTR*, *MC4R*, *GLP1R*, *PRLR*) the putamen. Numbers in receptor symbols reflect transcript abundances expressed as mean counts per million (cpms) from subjects #21 and 22. The figure illustrates receptors that were consistently observed in the given cell type of both human samples. INs: interneurons.

In contrast with the above suggestion, our results indicate that the human Pu and Cd contain mostly *bona fide* GnRH decapeptide because (1) human ChINs can be immunolabeled with the LR-1 antibodies and several other GAP1 and GnRH antibodies, unlike extrahypothalamic neurons of the monkey. (2) They possess the full enzyme set of GnRH biosynthesis, as revealed by deep transcriptome profiling. Finally, (3) human Pu extracts contain 3–4-times as much uncleaved GnRH decapeptide as GnRH1-5, as shown by results of HPLC-MS/MS studies. Different conclusion from the monkey and human studies may reflect species differences and the use of different methodological approaches.

We note that the human genome includes a fully functional *GNRH2* gene (*Stewart et al., 2009*), in addition to *GNRH1*. The GnRH signals we detected in the Pu are due to *GNRH1*, rather than *GNRH2* expression, because (1) extrahypothalamic GnRH-IR neurons also exhibit ISH signal for *GNRH1* mRNA, (2) they are IR to GAP1, which only has low homology with the corresponding GAP2 sequence, and (3) ChINs of the Pu express high levels of the *GNRH1* transcript, according to RNA-seq results.

Our data neither exclude nor support the possibility that the GnRH degradation product GnRH1-5 plays a role in the regulation of the human striatal neurocircuitry. Indeed, we found that GnRH1-5 was also detectable with HPLC-MS/MS in the *postmortem* Pu and Cd, albeit at ~70% lower levels than GnRH. Transcriptomic studies revealed that one of the putative GnRH1-5 receptors, *GPR101* (*Cho-Clark et al., 2014*), was expressed at high levels in SPNs (mean cpm: 58.4) and at low levels in ChINs (mean cpm: 6.4) (*Supplementary file 3*). This differential expression suggests that if GnRH1-5 is a physiological neurotransmitter of human ChINs, it may regulate SPNs via GPR101. We note that *THOP1* expression in ChINs is not a strong proof for the transmitter role of GnRH1-5. This endopeptidase has many substrates unrelated to GnRH degradation (*Orlowski et al., 1983*). Further, in RNA-seq studies, *THOP1* was expressed at higher levels in SPNs than in ChINs. Moreover, *bona fide* hypothalamic GnRH neurons in the developing monkey brain also exhibited THOP1 immunoreactivity (*Terasawa et al., 2001*).

## Overlap with cholinergic neurons and large cell numbers argue against the placodal origin

Total GnRH-IR cell numbers we calculated for the basal forebrain and the basal ganglia of three adult human brains (229,447, 155,357, and 104,699) exceeded all previous estimates of GnRH cell numbers in any species. For technical reasons, our calculation could even underestimate the real number of GnRH neurons. *Perimortem* and *postmortem* conditions could be suboptimal in some samples to achieve maximal detection sensitivity. In addition, the use of 100-μm-thick sections for IHC could also compromise the detection of low signal levels via reducing antibody penetration. Rance and coworkers identified 5800 type III GnRH neurons in the human basal forebrain complex rostral to the mammillary bodies, caudal to the optic chiasm, and ventral to the anterior commissure (*Rance et al., 1994*). This low cell number may reflect that tissues with these anatomical guidelines are devoid of the bulk of the Pu, which contained the majority (82%) of the extrahypothalamic GnRH neurons in our study.

Our initial hypothesis was that extrahypothalamic GnRH neurons are derived from the ~8000 GnRH neurons reported recently along the dorsal migratory route during embryonic/fetal development (*Casoni et al., 2016*). The placodal origin now seems to be very unlikely considering the much larger GnRH cell numbers we observed in the adult brain.

Extrahypothalamic GnRH neurons of the human central nervous system also appear to be homologous to the 'early type' GnRH neurons reported in the developing monkey embryo and fetus (*Quanbeck et al., 1997*). In this species, the 'early' and 'late' types of GnRH neurons were distinguished based on differences in their time of appearance, morphology, and immunoreactivity pattern using GnRH antibodies against different GnRH epitopes (*Quanbeck et al., 1997*; *Terasawa et al., 2001*). It was speculated that early GnRH neurons originated from the dorsal olfactory placode before olfactory pit formation at E30, migrated into the brain along the olfactory nerve, and settled in striatal and limbic structures of the fetal brain (*Quanbeck et al., 1997*). However, in a subsequent study these authors noted that a 10- to 10,000-fold increase in the number of 'early' GnRH neurons in the basal forebrain within just a few days indicates that early GnRH neurons might rather be derived from the ventricular wall of the telencephalic vesicle (*Terasawa et al., 2001*). The possibility of non-placodal GnRH neuron development is compatible with the in vitro capability of

hypothalamic and hippocampal progenitors to generate GnRH cells and all other neuroendocrine cell types (*Markakis et al., 2004*).

It is worth noting that our RNA-seq studies provided transcriptomic information about a mixed ChIN population of the Pu, whereas ChINs exhibit substantial diversity in their physiology, morphology, and connectivity (*Gonzales and Smith, 2015*). Subclasses differ in their developmental origin (medial ganglionic eminence, septal epithelium, or preoptic area) and transcription factor profiles (*Ahmed et al., 2019*). It is unknown which ChIN subpopulation expresses *GNRH1* and *GNRHR1* because selective harvesting of well-preserved cellular RNA specifically from the GnRH-IR ChINs is currently unresolved. Development of an 'immuno-LCM/RNA-seq' method would also allow us to compare the gene expression profiles of hypothalamic and extrahypothalamic GnRH neurons.

## Both hypothalamic and extrahypothalamic GnRH neurons use cholinergic co-transmission

ChAT co-expression provided evidence that extrahypothalamic GnRH neurons correspond to subpopulations of previously known cholinergic cells. These include ChINs of the Pu, which communicate locally with SPNs as well as projection neurons of the nbM, which innervate distant limbic structures (*Ahmed et al., 2019*). Although ChAT emerged as a hallmark of the extrahypothalamic GnRH system, we found evidence that a relatively large subset of human hypothalamic GnRH neurons also express the cholinergic marker enzyme. To our knowledge, this colocalization phenomenon has not been reported in any other species before. Studies on GW11 human fetuses established that migratory GnRH neurons in the nasal compartment are not cholinergic, whereas both hypothalamic and extrahypothalamic GnRH neurons already express the ChAT signal at this age.

Rodent GnRH neurons are regulated by cholinergic afferents but not known to co-express cholinergic markers (*Turi et al., 2008*). To address the possibility that the cholinergic phenotype of murine GnRH neurons has only been overlooked in previous studies, we have recently tried to detect ChAT immunoreactivity in GnRH neurons of the mouse preoptic area. The lack of success of this attempt (Skrapits et al., unpublished) suggests that the cholinergic phenotype of GnRH neurons is a true species difference between humans and rodents.

Laser microdissection of size-selected ChINs and SPNs is a highly efficient approach for transcriptome profiling of the two cell types from the *postmortem* brain.

Deep transcriptome profiling of *postmortem* human neurons is technically challenging. Difficulties include (1) compromised RNA quality, (2) lack of obvious marker signals to distinguish cell types, and (3) low RNA yield from the LCM-isolated 300–600 neurons. Our strategy to isolate size-selected ChINs and SPNs with LCM was justified by the RNA-seq results, which showed high enrichment of known cell type-specific marker genes in the two cell pools and millions of reads in each. As one-third of ChINs in the Pu also contain GnRH, deep transcriptome profiling of ChINs offered an insight into the extrahypothalamic GnRH neuron transcriptome. Although it was beyond the focus of our study, RNA-sequencing of ChINs and SPNs also unveiled the neurotransmitter and receptor profiles of the two cell types and provided information about the putative molecular interactions taking place in the Pu. The transcriptome databases allowed us to propose putative peptidergic mechanisms and, thus, build the partial molecular connectome model of ChINs and SPNs.

## GnRH acts outside the hypothalamus to regulate various reproductive and non-reproductive functions

Clearly, the functions of GnRH are far from being restricted to the regulation of hypophysial gonadotropin secretion. Its receptor transcript, *GNRHR1,* is expressed in normal peripheral endocrine tissues including the uterus, placenta, ovaries, testes, and prostate gland as well as in various tumor cell types (*Harrison et al., 2004*). High levels of *GNRHR1* mRNA and immunoreactivity were reported in pyramidal neurons of the human hippocampus and cerebral cortex (*Wilson et al., 2006*). GnRH analogues were anti-apoptotic in a rat model of ischemia/reperfusion (*Chu et al., 2010*). Further, GnRH increased hippocampal estradiol levels and the spontaneous firing and *GNRHR1* expression of pyramidal neurons and prevented memory deficits caused by amyloid β deposition (*Marbouti et al., 2020*). While the source of GnRH acting on hippocampal neurons remains to be explored, results of our studies indicate that GnRHR1 in ChINs of the basal ganglia can bind locally synthesized GnRH neuropeptide. ChINs of the striatum contribute as interneurons to the regulation

of cortico-striato-thalamocortical neural pathways. Functions associated with this circuitry include motor control, learning, language, reward, cognitive function, and addiction (*Fazl and Fleisher, 2018*). The exact role of GnRH/GnRHR1 signaling in these functions requires clarification. We note that various GABAergic interneurons of the Pu not studied here may also express GnRHR1 to serve as additional target cells for GnRH signaling by ChINs. Cholinergic neurons of the nbM, which project to the entire cortical mantle, olfactory tubercle, and amygdala, have been implicated in the control of attention, maintenance of arousal, and learning and memory formation (*Koulousakis et al., 2019*). It remains to be determined if GnRH synthesized by these neurons binds to postsynaptic target cells or acts on autoreceptors, as we proposed for striatal ChINs.

## Receptor profile of human cholinergic interneurons may offer new therapeutic targets to treat neurodegenerative disorders

In the absence of animal models, it is currently difficult to estimate the role and importance of GnRH/GnRHR1 signaling in the human basal ganglia and basal forebrain. Non-reproductive dysfunctions have not been characterized in GnRH-deficient patients (*Chan, 2011*) or in the more common cases of GnRHR1 deficiency (*Chevrier et al., 2011*; *Seminara et al., 1998*).

Future studies will need to clarify alterations of extrahypothalamic GnRH/GnRHR1 signaling in neurodegenerative disorders affecting various cholinergic systems. Leading symptoms and cognitive decline in Alzheimer's disease are due to the loss of basal forebrain cholinergic neurons many of which exhibited GnRH immunoreactivity in nbM. Parkinson's disease (PD) is characterized by motor symptoms such as abnormal involuntary movements, bradykinesia, rigidity, gait, and tremor. Non-motor symptoms often include cognitive impairment, mood disorders, sleep alterations, dysautonomia, anosmia, and hallucinations (*Perez-Lloret and Barrantes, 2016*; *Tubert and Murer, 2021*). Many of these malfunctions in PD can be explained with the loss of the nigrostriatal dopaminergic input and ameliorated with levodopa. However, gait disorders and cognitive impairment/dementia are most often unresponsive to dopamine precursor treatment. These data indicate the involvement of other neurotransmitter systems. In particular, loss of striatal dopamine input causes a local hypercholinergic state in the striatum with consequences reviewed recently (*Tubert and Murer, 2021*). This hypercholinergic state explains the success of early PD therapies with *Atropa belladonna* derivatives (*Goetz, 2011*). Although the low efficacy of anticholinergic drugs compared to levodopa and unwanted side effects limit the use of general anticholinergic strategies (*Katzenschlager et al., 2003*), selective inhibition of striatal ChINs has been proposed recently as a more promising strategy to improve the transmitter balance in dopamine-deprived basal ganglia (*Mallet et al., 2019*; *Tubert and Murer, 2021*). An important physiological mechanism to inhibit acetylcholine release from ChINs is via M2-type (M2 and M4) muscarinic autoreceptors coupled to $G_i$ proteins. Accordingly, deletion of M2-type autoreceptors results in increased striatal acetylcholine release (*Bonsi et al., 2008*). Autoinhibitory mechanism by muscarinic autoreceptors was found to be lost in PD animal models (*Ding et al., 2006*). Indeed, our RNA-seq analysis confirmed that human ChINs contain very high levels of *CHRM2* autoreceptors (*Figure 4B* and *Supplementary file 3*). We note that although the reproductive side effects would limit the use of GnRH analogues in clinical practice (*Almeida et al., 2004*), the transcriptome profile of ChINs (*Figure 5* and *Supplementary file 3*) offers a few alternative receptorial mechanisms to counteract the hyperactivity of ChINs in PD.

## Conclusions

This study reports the discovery and characterization of 150,000–200,000 GnRH-IR neurons, which are located in the basal ganglia and basal forebrain of the adult human brain. These extrahypothalamic GnRH cells represent subsets of previously known cholinergic neurons that mainly synthesize *bona fide* GnRH decapeptide. Unexpectedly, a large subpopulation of human hypothalamic GnRH neurons share this cholinergic (ChAT) neurochemistry, which has not been detected in rodents. RNA-seq experiments on ChINs and SPNs of the human Pu reveal that ChINs express *GNRH1* and *GNRHR1*, whereas their main target cells, the SPNs, do not, making it likely that GnRH acts via autoreceptors. The role of GnRH/GnRHR1 signaling within extrahypothalamic neuronal circuitries of the human brain requires clarification. RNA-seq studies, which revealed the transcriptome profiles of ChINs and SPNs, also provide an insight into the molecular connectome of the human putamen.

# Materials and methods

## Key resources table

| Reagent type (species) or resource | Designation | Source or reference | Identifiers | Additional information |
|---|---|---|---|---|
| Biological sample (*Homo sapiens*) | Striatum, hypothalamus, putamen, n. caudatus, claustrum (adult) | 1st Department of Pathology and Experimental Cancer Research, Semmelweis University, Budapest, Hungary | See *Supplementary file 1* | |
| Biological sample (*Homo sapiens*) | Head (fetus) | Agence de la Biomédecine, Saint-Denis la Plaine, France, protocol n°: PFS16–002 | See *Supplementary file 1* | |
| Antibody | Anti-GAP (rabbit polyclonal) | Dr. M. D. Culler *Culler and Negro-Vilar, 1986* | MC-2 | IHC (1:5000) |
| Antibody | Anti-GnRH (rabbit polyclonal) | Dr. R. A. Benoit *Silverman et al., 1990* | LR-1 | IHC (1:10,000) |
| Antibody | Anti-GnRH (guinea pig polyclonal) | Made in-house *Hrabovszky et al., 2011* | #1018 | IHC, IF-TSA (1:30,000) IF (1:10,000) |
| Antibody | Anti-GnRH (rat polyclonal) | Made in-house *Skrapits et al., 2015* | #1014 | IHC, IF-TSA (1:20,000) |
| Antibody | Anti-GnRH (sheep polyclonal) | Made in-house *Skrapits et al., 2015* | #2000 | IHC (1:1000) |
| Antibody | Anti-ChAT (goat polyclonal) | Merck | Cat#AB144P; RRID:AB_2079751 | IF (1:150), IF-TSA (1:2000) |
| Antibody | Anti-guinea pig IgG (H + L) Alexa Fluor 488 (donkey polyclonal) | Jackson ImmunoResearch | Cat#706-545-148; RRID:AB_2340472 | IF (1:400) |
| Antibody | Anti-goat IgG (H + L) Alexa Fluor 568 (donkey polyclonal) | Invitrogen | Cat#A-11057; RRID:AB_2534104 | IF (1:400) |
| Antibody | Biotin-SP-AffiniPure anti-guinea pig IgG (H + L) (donkey polyclonal) | Jackson ImmunoResearch | Cat#706-065-148; RRID:AB_2340451 | IHC (1:500) |
| Antibody | Biotin-SP-AffiniPure anti-rabbit IgG (H + L) (donkey polyclonal) | Jackson ImmunoResearch | Cat#711-065-152; RRID:AB_2340593 | IHC (1:500) |
| Antibody | Biotin-SP-AffiniPure anti-rat IgG (H + L) (donkey polyclonal) | Jackson ImmunoResearch | Cat#712-065-153; RRID:AB_2315779 | IHC (1:500), IF-TSA (1:500) |
| Antibody | Biotin-SP-AffiniPure anti-sheep IgG (H + L) (donkey polyclonal) | Jackson ImmunoResearch | Cat#713-065-147; RRID:AB_2340716 | IHC (1:500) |
| Antibody | Biotin-SP-AffiniPure anti-goat IgG (H + L) (donkey polyclonal) | Jackson ImmunoResearch | Cat#705-065-147; RRID:AB_2340397 | IF-TSA (1:500) |
| Antibody | Peroxidase-AffiniPure anti-guinea pig IgG (H + L) (donkey polyclonal) | Jackson ImmunoResearch | Cat#706-035-148; RRID:AB_2340447 | IF-TSA (1:250) |
| Antibody | Peroxidase-conjugated anti-digoxigenin, Fab fragments (sheep polyclonal) | Roche | 11207733910; RRID:AB_514500 | ISH (1:100) |

*Continued on next page*

*Continued*

| Reagent type (species) or resource | Designation | Source or reference | Identifiers | Additional information |
|---|---|---|---|---|
| Other | FITC-tyramide | Synthesized in-house *Hopman et al., 1998* | | IF-TSA (1:1000) |
| Other | Cy3-tyramide | Synthesized in-house *Hopman et al., 1998* | | IF-TSA, ISH (1:1000) |
| Other | ABC elite reagent | Vector Laboratories | PK-6100; RRID:AB_2336819 | IHC, IF-TSA (1:1000) |
| Sequence-based reagent | Antisense proGnRH | This paper | ISH probe | CATTCACAACACAGCA CTTTATTATGGAATATG TGCAACTTGGTGTAAGG ATTTCTGAAATTCATACC ATTTACAGGTATTTAATG GGTTATAAATTTTCAATG TCAGAATTATACTTAAGT CATGTTAGTAATGGTCAT TCCTTCTGGCCCAATGG ATTTAAATCTTCTTCTGC CCAGTTTCCTCTTCAATC AGACTTTCCAGAGCTCC TTTCAGGTCTCGGAGGG GAGAACGTGGCTGGTGC GTGGTGCATTCGAAGCG TTGGGTTTCTGCCAGTT GACCAACCTCTTTGACT ATCTCTTGGAAAGAATCA ATCAAATTTTCGGCATCT CTCTTTCCTCCAGGGCG CAGTCCATAGGACCAGT GCTGGCTGGAGCAGCCT TCCACGCACCAAGTCAG TAGAATAAGGCCAGCTA GGAGTTTTTGAATTGGC TTCATTCTGTTTAGAGG CAGAGAGCCAAAAAGATCC |
| Peptide, recombinant protein | GnRH decapeptide | Merck | Cat#L8008 | |
| Commercial assay or kit | Arcturus Paradise Plus RNA Extraction and Isolation Kit | ThermoFisher | Cat#KIT0312I | |
| Commercial assay or kit | RNA 6000 Pico Kit | Agilent | 5067-1513 | |
| Commercial assay or kit | TruSeq Stranded Total RNA Library Preparation Gold Kit | Illumina | Cat#20020598 | |
| Commercial assay or kit | High Sensitivity DNA Kit | Agilent | 5067-4626 | |
| Commercial assay or kit | NextSeq 500/550 High Output (v2.5) Kit | Illumina | Cat#20024906 | |
| Software, algorithm | AxioVision Imaging System 4.6 | Carl Zeiss | RRID:SCR_002677 | |
| Software, algorithm | Zen Black v.14.0.12.201 | Carl Zeiss | RRID:SCR_018163 | |
| Software, algorithm | PALM MicroBeam | Carl Zeiss | RRID:SCR_020929 | |
| Software, algorithm | Agilent Bioanalyzer 2100 Expert | Agilent Technologies | RRID:SCR_018043 | |

## Human subjects

Adult human brain tissues from male and female individuals without known neurological disorders were collected from autopsies (N = 28) at the 1st Department of Pathology and Experimental Cancer Research, Semmelweis University, Budapest, Hungary. Quantitative analyses were performed on tissues from several subjects to compensate for unavoidable biological (sex, age, health status of human subjects) and methodological (*perimortem* conditions and *postmortem* time) differences among the samples. Ethic permissions were obtained from the Regional and Institutional Committee of Science and Research Ethics of Semmelweis University (SE-TUKEB 251/2016), in accordance with the Hungarian Law (1997 CLIV and 18/1998/XII.27. EÜM Decree/) and the World Medical Association Declaration of Helsinki. The demographic data of donors and use of their tissue samples in the different experiments are summarized in *Supplementary file 1*, whereas the most important technical details of IHC studies are presented in *Supplementary file 2*. The dissected brain tissue blocks were rinsed briefly with running tap water. Then, depending on use, they were either immersion-fixed with buffered paraformaldehyde (PFA) as detailed below or snap-frozen on powdered dry ice.

## Human fetuses

Fetal tissues (#29 and 30; *Supplementary file 1*) were made available in accordance with French bylaws (Good Practice Concerning the Conservation, Transformation, and Transportation of Human Tissue to Be Used Therapeutically, published on December 29, 1998). The studies on human fetal tissue were approved by the French agency for biomedical research (Agence de la Biomédecine, Saint-Denis la Plaine, France, protocol no.: PFS16-002). Non-pathological human fetuses were obtained at GW11 from pregnancies terminated voluntarily after written informed consent of the parents (Gynaecology Department, Jeanne de Flandre Hospital, Lille, France).

## Mapping and quantitative analysis of extrahypothalamic GnRH neurons in adult brains

Three brains (#1–3) were cut into ~15-mm-thick coronal slices. The tissue slabs were immersion-fixed in several changes of buffered (0.1 M PBS; pH 7.4) 4% PFA for 21 days and then infiltrated with 20% sucrose for 7 days (4°C). The right hemispheres were isolated and processed to determine the distribution and number of extrahypothalamic GnRH neurons in the nucleus caudatus (Cd), putamen (Pu), globus pallidus (GP), nucleus accumbens (nAcc), bed nucleus of the stria terminalis (BnST), and nucleus basalis of Meynert (nbM). Brain slices were embedded in Jung tissue freezing medium (Leica Biosystems, Nussloch, Germany), snap-frozen on powdered dry ice. Then, 100-μm-thick coronal sections were collected with a Leica SM 2000R freezing microtome into tissue culture plates filled with anti-freeze solution (30% ethylene glycol, 25% glycerol, 0.05 M phosphate buffer, pH 7.4) and stored at −20°C. Every 24th section between Bregma levels −22.5 and 33.1 (*Mai et al., 1997*) was immunostained using a well-characterized guinea pig antiserum (#1018) against GnRH decapeptide (*Hrabovszky et al., 2011*; *Figure 1A*). The sections were rinsed in PBS, pretreated with a mixture of 1% $H_2O_2$ and 0.5% Triton X-100 for 30 min, and subjected to antigen retrieval with 0.1 M citrate buffer (pH 6.0) at 80°C for 30 min. To maximize signal, immunohistochemical incubations were extended: guinea pig anti-GnRH antibodies (#1018; 1:30,000) (*Hrabovszky et al., 2011*), 5 days; biotinylated donkey anti-guinea pig IgG antibodies (Jackson ImmunoResearch Europe, Cambridgeshire, UK; 1:500), 12 hr; ABC Elite reagent (Vector, Burlingame, CA; 1:1000), 4 hr. The signal was visualized with nickel-diaminobenzidine (Ni-DAB) chromogen (10 mg diaminobenzidine, 30 mg nickel-ammonium-sulfate, and 0.003% $H_2O_2$ in 20 ml Tris-HCl buffer solution [0.05 M; pH 8.0]). Immunostained sections were mounted on 75 mm × 50 mm microscope slides from 0.3% polyvinyl alcohol, air-dried, dehydrated with 70, 95, and 100% ethanol (5 min each), cleared with xylenes (2 × 5 min), and coverslipped with DPX mounting medium (Merck, Darmstadt, Germany).

Anatomical sites to be analyzed separately were identified at each rostro-caudal level (*Mai et al., 1997*) by macroscopic and microscopic analyses and their borders were marked on the coverslips. Labeled cell bodies were counted in each region with light microscopy, and cell numbers were corrected against overcounting (*Figure 1—figure supplement 1A*) using Abercrombie's correction factor T/(T + h), where T is actual section thickness and h is the average diameter of GnRH neurons along the Z axis (*Guillery, 2002*). Two Pu sections were used to determine T and h (#2, 3). These sections were processed for the IF detection of GnRH neurons with guinea pig anti-GnRH antibodies

(#1018; 1:30,000; 5 days), followed by peroxidase-conjugated anti-guinea pig IgG (Jackson ImmunoResearch; 1:250; 4 hr) and FITC-tyramide (*Hopman et al., 1998*) (diluted 1:1000 with 0.05 M Tris-HCl buffer/0.005% $H_2O_2$, pH 7.6, 30 min). The sections were embedded into 2% agarose and resectioned with a Leica vibratome perpendicularly to the original section plane. T and h were measured with confocal microscopy to calculate a final correction factor of 0.712 (*Figure 1—figure supplement 1B*). The number of GnRH cells (n) counted in every 24th section of a single hemisphere was first doubled (with the assumption that the distribution of extrahypothalamic GnRH neurons is symmetrical) and then multiplied by 24 and Abercrombie's correction factor to obtain an estimate of the total number of extrahypothalamic GnRH neurons ($\Sigma = n \times 2 \times 24 \times 0.712$) in the basal ganglia and basal forebrain of each brain.

## Immuno-peroxidase detection of extrahypothalamic GnRH neurons using different primary antibodies as positive controls

Dissected tissue samples (N = 10; #5, 6, and 12–19) containing the extrahypothalamic regions of interest were fixed by immersion in freshly prepared 4% PFA in PBS for 14–21 days at 4°C. The fixed blocks were trimmed, infiltrated with 20% sucrose for 5 days at 4°C, placed in a freezing mold, surrounded with Jung tissue freezing medium, snap-frozen on powdered dry ice, and sectioned coronally at 20–30 μm with a freezing microtome (Leica Biosystems). The sections were stored permanently in anti-freeze solution (30% ethylene glycol, 25% glycerol, 0.05 M phosphate buffer, pH 7.4) at −20°C. Following the pretreatments detailed above, a series of different GnRH and GAP1 antibodies (*Supplementary file 2*) were tested for reactivity with extrahypothalamic GnRH neurons. These included the guinea pig (#1018; 1:30,000) (*Hrabovszky et al., 2011*), rat (#1044; 1:20,000) (*Skrapits et al., 2015*), and sheep (#2000; 1:1000) (*Skrapits et al., 2015*) polyclonal antisera generated in our laboratory against the GnRH decapeptide and the LR-1 rabbit GnRH antiserum (1:10,000; gift from Dr. R. A. Benoit), which was reported not to produce specific labeling of extrahypothalamic GnRH neurons in embryonic/fetal rhesus monkeys (*Quanbeck et al., 1997*; *Terasawa et al., 2001*). In addition, a rabbit polyclonal antiserum (MC-2; 1:5000) (*Culler and Negro-Vilar, 1986*) to aa 25–53 of hGAP1 (accession: P01148) was used. Following a 48 hr incubation in primary antibodies (4°C), the signals were detected using biotinylated secondary antibodies (Jackson ImmunoResearch; 1:500; 1 hr), ABC Elite reagent (Vector; 1:1000; 1 hr), and Ni-DAB chromogen. The immunostained sections were coverslipped with DPX.

## Preabsorption control with the guinea pig GnRH antiserum (#1018)

The 1:30,000 working solution of the guinea pig polyclonal GnRH antiserum (#1018) was preabsorbed overnight with 0.1 μg/ml GnRH decapeptide at 4°C. Test sections of the Pu (N = 3; #17–19) were incubated in preabsorbed and control antisera for 48 hr at 4°C, and then processed in parallel for the immuno-peroxidase detection of GnRH as described above.

## Dual-label immunofluorescence experiments used as a positive control for GnRH labeling

Positive control experiments with IF double-labeling used two sequential rounds of tyramide signal amplification (TSA) to maximize both GnRH signals. Sections from two subjects (#4 and 5) were pretreated as above, followed by an additional Sudan Black step (*Mihály et al., 2000*) to quench tissue autofluorescence. Then, a mixture of guinea pig GnRH (#1018; 1:30,000) and rat GnRH (#1044; 1:20,000) primary antibodies was applied to the sections for 48 hr at 4°C, followed by peroxidase-conjugated anti-guinea pig IgG (Jackson ImmunoResearch; 1:250; 1 hr) and Cy3-tyramide (*Hopman et al., 1998*) (diluted 1:1000 with 0.05 M Tris-HCl buffer/0.005% $H_2O_2$, pH 7.6, 30 min). Peroxidase was inactivated with 0.5% $H_2O_2$ and 0.1 M sodium azide in PBS for 30 min. Then, the rat GnRH antibodies were reacted with biotin-conjugated secondary antibodies (Jackson ImmunoResearch; 1:500; 1 hr), ABC Elite reagent (Vector; 1:1000; 1 hr), and FITC-tyramide (*Hopman et al., 1998*) (diluted 1:1000 with 0.05 M Tris-HCl buffer/0.005% $H_2O_2$, pH 7.6, 30 min). The dual-labeled sections were mounted and coverslipped with the aqueous mounting medium Mowiol.

## In situ hybridization detection of *GNRH1* mRNA in GnRH neurons of the human putamen

The digoxigenin-labeled antisense probe targeting bases 32–500 of human *GNRH1* mRNA (NM_001083111.2) was transcribed in the presence of digoxigenin-11-UTP (Merck Millipore) in a reaction mixture containing linearized cDNA template (1 µg), 5× transcription buffer (2 µl), 100 mM DTT (1 µl), 10 mM ATP, CTP, and GTP (0.5 µl each), 10 mM digoxigenin-11-UTP (0.5 µl), 1 mM UTP (1 µl), 40 U/µl RNase inhibitor (RNasin; Promega, Madison, WI; 0.5 µl), and 20 U SP6 RNA polymerase (Promega; 1 µl). Following a 1 hr incubation of the cocktail at 37°C, a second 20 U aliquot of SP6 RNA polymerase was added and the reaction was allowed to proceed for an additional 1 hr. The volume was brought up to 90 µl with nuclease-free water, and the cDNA template was digested for 30 min at 37°C after the addition of 1 µl DNase I (10 U/µl; Roche Diagnostics, Rotkreuz, Switzerland), 5 µl 1 M Tris/HCl buffer (pH 8.0), 1 µl transfer RNA (tRNA; 25 mg/ml), 1 µl 1 M $MgCl_2$, and 0.5 µl RNasin (40 U/µl) to the reaction mixture. The cRNA probe was purified using sodium chloride/ethanol precipitation, dissolved in 100 µl of 0.1% sodium dodecyl sulfate, stored at −20°C, and added to the hybridization buffer (50% formamide, 2× SSC, 20% dextran sulfate, 1× Denhardt's solution, 500 µg/ml yeast tRNA, 50 mM DTT) at a 1:100 dilution (1× SSC = 0.15 M NaCl/0.015 M sodium citrate, pH 7.0). 4-mm-thick putamen blocks were dissected out from five brains (#15–19), immersion-fixed in 4% PFA for 48 hr, and infiltrated with 20% sucrose for 48 hr. 20-µm-thick floated sections were prepared with a freezing microtome and processed for combined ISH detection of *GNRH1* mRNA and IF detection of GnRH peptide. First, the sections were acetylated with 0.25% acetic anhydride in 0.9% NaCl/0.1 M triethanolamine-HCl for 10 min, rinsed in 2× SSC for 2 min, treated sequentially with 50, 70, and 50% acetone (5 min each), rinsed with 2× SSC, and hybridized overnight in microcentrifuge tubes containing the hybridization solution. Non-specifically bound probes were digested with 20 µg/ml ribonuclease A (Merck; dissolved in 0.5 M NaCl/10 mM Tris-HCl/1 mM EDTA; pH 7.8) for 60 min at 37°C, followed by a 60-min-stringent treatment (55°C in 0.1× SSC solution) to reduce background. The floated sections were rinsed briefly with 100 mM maleate buffer (pH 7.5) and blocked for 30 min against non-specific antibody binding with 2% blocking reagent (Merck) in maleate buffer. To detect the hybridization signal, the sections were incubated overnight at 4°C in digoxigenin antibodies conjugated to horseradish peroxidase (anti-digoxigenin-POD; Fab fragment; 1:100; Roche), rinsed in TBS (0.1 M Tris-HCl with 0.9% NaCl, pH 7.8), and then reacted with FITC-tyramide (*Hopman et al., 1998*) (diluted 1:1000 with 0.05 M Tris-HCl buffer/0.005% $H_2O_2$; pH 7.6) for 30 min. Peroxidase was inactivated with 0.5% $H_2O_2$ and 0.1 M sodium azide in PBS for 30 min. Subsequently, GnRH immunoreactivity was detected with guinea pig anti-GnRH (#1018; 1:30,000) primary antibodies (48 hr at 4°C), biotin-conjugated secondary antibodies (Jackson ImmunoResearch; 1:500; 1 hr), ABC Elite reagent (Vector; 1:1000; 1 hr), and Cy3-tyramide (*Hopman et al., 1998*) (diluted 1:1000 with 0.05 M Tris-HCl buffer/0.005% $H_2O_2$, pH 7.6, 30 min).

## DiI labeling of putamen sections to study GnRH cell morphology

Combined IF detection of peptidergic neurons and their Golgi-like cell membrane labeling with the lipophilic dye DiI using a Gene Gun was adapted to studies of human extrahypothalamic GnRH neurons from our recently reported procedure (*Takács et al., 2018*). A 4-mm-thick tissue block was dissected out from the Pu of a 72-year-old female subject (#20) and immersion-fixed lightly with freshly prepared 2% PFA in 0.1 M PBS (pH 7.4) for 14 days (4°C). 100-µm-thick coronal sections were prepared with a Leica VTS-1000 Vibratome (Leica Biosystems) and stored in PBS/0.1% sodium azide at 4°C before use. The sections were pretreated with a mixture of 1% $H_2O_2$ and 0.5% Tween 20 for 30 min, followed by epitope retrieval with 0.1 M citrate buffer (pH 6.0) at 80°C for 30 min. Then, sequential incubations were carried out in the guinea pig GnRH antibodies (#1018; 1:30,000) for 5 days, peroxidase-conjugated anti-guinea pig antibodies (Jackson ImmunoResearch Laboratories; 1:250) for 4 hr, and finally, FITC-tyramide (diluted 1:1000 with 0.05 M Tris-HCl buffer/0.005% $H_2O_2$, pH 7.6, 30 min) prepared (*Hopman et al., 1998*) and used (*Takács et al., 2018*) as reported. Methods to prepare and deliver DiI-coated tungsten particles with a Helios Gene Gun (Bio-Rad, Hercules, CA) were adapted from published procedures (*Seabold et al., 2010*; *Staffend and Meisel, 2011*). Sections of the Pu were transferred into 12-well tissue culture plates containing PBS. The buffer was removed with a pipette and diolistic labeling was carried out using a 40 mm spacer and a 120–150 pounds per square inch (PSI) helium pressure, which resulted in random labeling of cells, including

12 GnRH-IR neurons. Labeled sections were rinsed in PBS/0.1% sodium azide/0.2% EDTA and the lipophilic dye was allowed to diffuse along the cytoplasmic membranes for 24 hr at 4°C. The sections were coverslipped with Mowiol to study the Golgi-like DiI labeling of the randomly hit GnRH neurons.

## Dual-label immunofluorescence experiments to colocalize choline acetyltransferase with GnRH

Sections from striatal (N = 4; #3–5 and 19) and hypothalamic (N = 7; #3 and 6–11) samples were rinsed in PBS followed by a mixture of 1% $H_2O_2$ and 0.5% Triton X-100 for 30 min, and then subjected to antigen retrieval in 0.1 M citrate buffer (pH = 6.0) at 80°C for 30 min and Sudan Black pretreatment. GnRH neurons were detected using sequentially guinea pig GnRH antibodies (#1018; 1:30,000; 48 hr; 4°C), peroxidase-conjugated anti-guinea pig IgG (Jackson ImmunoResearch; 1:250; 1 hr), and FITC-tyramide (*Hopman et al., 1998*) (diluted 1:1000 with 0.05 M Tris-HCl buffer/0.005% $H_2O_2$, pH 7.6, 30 min). Peroxidase was inactivated with 0.5% $H_2O_2$ and 0.1 M sodium azide in PBS for 30 min. Then, ChAT neurons were detected using goat anti-ChAT antibodies (AB144P; Merck; 1:2000) (*Yonehara et al., 2011*) for 48 hr at 4°C, followed by biotinylated secondary antibodies (donkey anti-goat IgG; Jackson ImmunoResearch; 1:500; 1 hr), ABC Elite reagent (Vector; 1:1000, 1 hr), and Cy3-tyramide (*Hopman et al., 1998*) (diluted 1:1000 with 0.05M Tris-HCl buffer, pH 7.6, containing 0.005% $H_2O_2$, 30 min). The dual-labeled sections were mounted on slides, coverslipped with Mowiol, and analyzed with confocal microscopy. Confocal Z-stacks were prepared from each region and analyzed to determine the percentage of GnRH neurons showing ChAT immunoreactivity and vice versa.

## Dual-immunofluorescence studies of fetal tissues

The fetuses (N = 2; #29 and 30) were fixed by immersion in 4% buffered PFA at 4°C for 5 days. The tissues were then cryoprotected in PBS containing 30% sucrose at 4°C overnight, embedded in Tissue-Tek OCT compound (Sakura, Finetek), frozen in dry ice, and stored at −80°C until sectioning. Frozen samples were cut serially at 20 µm with a cryostat (Leica Biosystems) and immunolabeled, as described previously (*Casoni et al., 2016*), with polyclonal goat anti-ChAT (AB144P; Merck; 1:150) and guinea pig anti-GnRH antibodies (#1018; 1:10,000), in a solution containing 10% normal donkey serum and 0.3% Triton X-100 at 4°C for 3 days. 3 × 10 min washes in 0.01 M PBS were followed by incubations in AF568-conjugated donkey anti-goat (Invitrogen; 1:400) and AF488-conjugated donkey anti-guinea pig (Jackson ImmunoResearch; 1:400) antibodies for 1 hr each. The sections were counterstained with Hoechst (1:1000) and coverslipped with Mowiol.

## RNA-sequencing

### Reagents

For all experiments, nuclease-free water was used and reagents were of molecular biology grade. Work surfaces and equipment were cleaned with RNaseZAP.

### Section preparation

After dissection, tissue samples from the Pu of two subjects (#21 and 22) were snap-frozen in −40°C isopentane precooled with a mixture of dry ice and ethanol. Then, 20 µm-thick coronal sections were cut with a Leica CM1860 UV cryostat (Leica Biosystems. Wetzlar, Germany), collected onto PEN membrane glass slides (Membrane Slide 1.0 PEN, Carl Zeiss, Göttingen, Germany), air-dried for 5 min in the cryostat chamber, and fixed with a mixture of 2% PFA, 0.1% diethyl pyrocarbonate, 1% sodium acetate, and 70% ethanol (10 min). After brief rehydration (RNase-free water 2 min), sections were stained with 0.5% cresyl violet solution (1 min), rinsed in RNase-free water, and dehydrated again in 70, 96, and 100% ethanol (30 s each). The slides were kept at −80°C in clean slide mailers containing silica gel desiccants until further processing.

### Laser-capture microdissection

Slides were placed into the slide holder of the microscope, and 300 ChINs were microdissected by LCM using a PALM Microbeam system (Zeiss). The cells were pressure-catapulted from the object plane into 0.5 ml tube caps (Adhesive Cap 200, Zeiss) with a single laser pulse using a 40× objective

lens. A second control cell pool was prepared from 600 medium-sized neurons most of which corresponded to SPNs. The mean profile areas of ChINs and SPNs were 674.76 $\mu m^2$ and 161.22 $\mu m^2$, respectively. The LCM caps were stored at $-80°C$ until RNA extraction.

## RNA extraction, RNA-seq library preparation and sequencing

The Arcturus Paradise Plus RNA Extraction and Isolation Kit (ThermoFisher, Waltham, MA) was used to isolate total RNA according to the manufacturer's protocol. Samples collected from control sections of the two brains showed RNA integrity numbers (RINs) of 5.7 and 4.1, respectively, as determined using Bioanalyzer Eukaryotic Total RNA Pico Chips (Agilent, Santa Clara, CA). RNA samples were converted to RNA-seq libraries with the TruSeq Stranded Total RNA Library Preparation Gold kit (Illumina, San Diego, CA). This kit was reported to reliably and reproducibly generate libraries from 1 to 2 ng input RNA (*Schuierer et al., 2017*). The manufacturer's protocol was followed, except for the use of 16, instead of 15, cycles of amplification for adaptor-ligated DNA fragment enrichment. Single-end sequencing was performed on Illumina NextSeq500 instrument using the Illumina NextSeq500/550 High Output kit v2.5 (75 cycles).

## Bioinformatics

After quality check with FastQC, raw reads were cleaned by trimming low-quality bases by Trimmomatic 0.39 (settings: LEADING:3, TRAILING:3, SLIDINGWINDOW:4:30, MINLEN:50). The prepared reads were mapped to the GRCh38.p13 human reference genome using STAR (v 2.7.3a) (*Dobin et al., 2013*) with an average overall alignment rate of 68.4% (s.d. = 9.9%). Gene-level quantification of read counts based on human genome with Ensembl (release 99) (*Yates et al., 2020*) annotation was performed by featureCounts (subread v 2.0.0) (*Liao et al., 2014*), with a mean of 30.4% (s.d. = 9.7%) of mapped reads assigned to genes in the case of the four samples. The raw read counts per genes were normalized and processed further in R (R2020) with the package DESeq2 (*Love et al., 2014*) and edgeR (*McCarthy et al., 2012*). For feature annotation, the R package KEGGREST (Dan Tenenbaum, KEGGREST: Client-side REST access to KEGG. R package version 1.26.1; 2019) and the PANTHER database (v. 15.0) (*Thomas et al., 2003*) were used.

## High-performance liquid chromatography-tandem mass spectrometry

Brain tissue specimens (N = 8; #21–28) were snap-frozen and kept at $-80°C$. Approximately 10–60 mg samples were microdissected in a $-20°C$ cryostat chamber from the MBH (N = 2), Pu (N = 5), Cd (N = 2), and Cl (N = 3). After addition of the extraction solution containing 1% acetic acid and Complete Mini protease inhibitor cocktail (Roche, Basel, Switzerland) in 1:2 w/v proportion, samples were homogenized using an ultrasonic sonotrode. The homogenates were mixed with double volume acetonitrile and centrifuged to produce protein-free supernatants. Separation of 10 µl samples was carried out by HPLC (Perkin Elmer Series 200) using gradient elution on a Luna Omega Polar C18 50 × 3 mm, 3 µm column (Phenomenex, Torrance, CA). Acetonitrile and 0.1% formic acid were applied for gradient elution with the flow rate of 500 µl/min. Acetonitrile increased from 10% to 40% in 3 min, and this was maintained for 0.5 min. The initial 10% was reached in 0.5 min and maintained for 2 min. Analytes were detected using a triple quadrupole MDS SCIEX 4000 Q TRAP mass spectrometer (Applied Biosystems) in positive multiple reaction monitoring mode (MRM transitions: GnRH: 592.1 → 249.3, GnRH1-5: 671.2 → 159.1). Peak areas were integrated with Analyst 1.4.2 software (Sciex, Framingham, MA), and concentrations were calculated using matrix-matched calibration.

## Light microscopy

Representative light microscopic images were prepared with an AxioCam MRc 5 digital camera mounted on a Zeiss AxioImager M1 microscope using the AxioVision 4.6 software (Carl Zeiss, Göttingen, Germany).

## Confocal microscopy

Fluorescent signals were studied with a Zeiss LSM780 confocal microscope. High-resolution images were captured using a 20×/0.8 NA objective, a 0.6–1× optical zoom, and the Zen software (Carl Zeiss). Different fluorochromes were detected with laser lines 488 nm for FITC and AF488 and 561 nm for Cy3. Emission filters were 493–556 nm for FITC and AF488 and 570–624 nm for Cy3. To

prevent emission crosstalk between the fluorophores, the red channel was recorded separately from the green one ('smart setup' function). To illustrate the results, confocal Z-stacks (Z-steps: 0.85–1 μm; pixel dwell time: 0.79–1.58 μs; resolution: 1024 × 1024 pixels; pinhole size: set at 1 Airy unit) were merged using maximum intensity Z-projection (ImageJ). The final figures were adjusted in Adobe Photoshop using the magenta-green color combination and saved as TIF files.

Fetal sections were examined using an Axio Imager.Z1 ApoTome microscope (Carl Zeiss, Germany) equipped with a motorized stage and an AxioCam MRm camera (Zeiss). For confocal observation and analyses, an inverted laser scanning Axio observer microscope (LSM 710, Zeiss) with an EC Plan NeoFluorÅ ~100/1.4 numerical aperture oil-immersion objective (Zeiss) was used (Imaging Core Facility of IFR114, of the University of Lille, France).

## Acknowledgements

We thank the midwives of the Gynecology Department, Jeanne de Flandre Hospital of Lille (Centre d'Orthogénie), France, for their kind assistance and support; M Tardivel and A Bongiovanni (BICeL core facility of Lille, Univ. Lille, CNRS, Inserm, CHU Lille, Institut Pasteur de Lille, US 41-UMS 2014-PLBS, F-59000 Lille, France) for expert technical assistance. The authors acknowledge support of the Inserm Cross-Cutting Scientific Program (HuDeCA) and are grateful to Dr. R. A. Benoit for the LR-1 antiserum.

## Additional information

### Funding

| Funder | Grant reference number | Author |
|---|---|---|
| Hungarian Science Foundation | K128317 | Erik Hrabovszky |
| Hungarian Science Foundation | PD134837 | Katalin Skrapits |
| Hungarian Brain Research Program | 2017-1.2.1-NKP- 2017-00002 | Erik Hrabovszky |
| Institut National de la Santé et de la Recherche Médicale | U1172 | Vincent Prévot Paolo Giacobini |
| Agence Nationale de la Recherche | ANR-19-CE16-0021-02 | Paolo Giacobini |
| Inserm Cross-Cutting Scientific Program | HuDeCa | Paolo Giacobini |
| NRDI Office | TKP2020 IES | Blanka Tóth |
| NRDI Office | BME-IE-BIO | Blanka Tóth |

The funders had no role in study design, data collection and interpretation, or the decision to submit the work for publication.

### Author contributions

Katalin Skrapits, Paolo Giacobini, Erik Hrabovszky, Conceptualization, Supervision, Funding acquisition, Investigation, Methodology, Writing - original draft, Writing - review and editing; Miklós Sárvári, Éva Rumpler, Norbert Solymosi, Blanka Tóth, Conceptualization, Investigation, Methodology, Writing - original draft, Writing - review and editing; Imre Farkas, Investigation; Balázs Göcz, Szabolcs Takács, Conceptualization, Investigation, Methodology; Viktória Váczi, Cecile Allet, Ludovica Cotellessa, Investigation, Methodology; Csaba Vastagh, Conceptualization; Gergely Rácz, András Matolcsy, Ferenc Erdélyi, Gábor Szabó, Michael D Culler, Methodology; Szilárd Póliska, Conceptualization, Methodology; Vincent Prévot, Conceptualization, Supervision, Funding acquisition, Writing - original draft, Writing - review and editing

## Author ORCIDs

Imre Farkas [iD] https://orcid.org/0000-0002-0159-4408
Norbert Solymosi [iD] http://orcid.org/0000-0003-1783-2041
Paolo Giacobini [iD] http://orcid.org/0000-0002-3075-1441
Erik Hrabovszky [iD] https://orcid.org/0000-0001-6927-0015

## Ethics

Human subjects: Ethic permissions were obtained from the Regional and Institutional Committee of Science and Research Ethics of Semmelweis University (SE-TUKEB 251/2016), in accordance with the Hungarian Law (1997 CLIV and 18/1998/XII.27. EÜM Decree/) and the World Medical Association Declaration of Helsinki. As explicitly declared in the Hungarian Law on Healthcare (1997. CLIV), tissue removal for either donation or research purposes can be done legally in Hungary, unless the deceased person banned the removal in advance. Therefore, collection and use of adult brain samples in this study did not require a priori informed consent of the deceased. The above cited legal article has been extended in 18/1998. (XII. 27. EÜM rendelet) decree of the Hungarian Ministry for Healthcare  http://net.jogtar.hu/jr/gen/hjegy_doc.cgi?docid=99800018.EUM. Paragraph 6. speaks about requirements for "Removal of organ or tissue from a cadaver". Paragraph 8. states: Before removal the healthcare professional must check whether the personal documents or health care documentation of deceased person contains a "Declaration of Objection", i.e. explicit statement that he/she does not allow the removal of tissue or organs. In paragraph 8. (3): If such declaration does not exist, then the tissue or organs can be removed. The later Hungarian Law 1999. LXXI. also addresses this issue in the very same spirit: https://mkogy.jogtar.hu/jogszabaly?docid=99900071.TV Fetal tissues were made available in accordance with French bylaws (Good Practice Concerning the Conservation, Transformation, and Transportation of Human Tissue to Be Used Therapeutically, published on December 29, 1998). The studies on human fetal tissue were approved by the French agency for biomedical research (Agence de la Biomédecine, Saint-Denis la Plaine, France, protocol n°: PFS16-002). Non-pathological human fetuses were obtained at gestational week 11 from pregnancies terminated voluntarily after written informed consent of the parents (Gynaecology Department, Jeanne de Flandre Hospital, Lille, France).

Animal experimentation: Experiments involving genetically modified male mice were carried out in accordance with the Institutional Ethical Codex, Hungarian Act of Animal Care and Experimentation (1998, XXVIII, section 243/1998) and the European Union guidelines (directive 2010/63/EU), and with the approval of the Institutional Animal Care and Use Committee of the Institute of Experimental Medicine. All measures were taken to minimize potential stress or suffering during sacrifice and to reduce the number of animals to be used.

## Decision letter and Author response

Decision letter https://doi.org/10.7554/eLife.67714.sa1
Author response https://doi.org/10.7554/eLife.67714.sa2

# Additional files

## Supplementary files

• Supplementary file 1. Demographic information about the 30 donors and use of tissue specimens in different experiments. ChAT: choline acetyltransferase; ChINs: cholinergic interneurons; GW11: gestational week 11; IF: immunofluorescence; IHC: immunohistochemistry; ISH: in situ hybridization; HPLC-MS/MS: high-performance liquid chromatography-tandem mass spectrometry; PMI: *postmortem* interval; RIN: RNA integrity number; RNA-seq: RNA sequencing; SPNs: medium spiny projection neurons.

• Supplementary file 2. Basic data on antibody specification, concentration, previous characterization, and on immunohistochemical reagents and applied detection methods. ChAT: choline acetyltransferase; GAP1: GnRH-associated peptide-1; GnRH: gonadotropin-releasing hormone; IF: immunofluorescence; IHC: immunohistochemistry; TSA: tyramide signal amplification.

• Supplementary file 3. Detailed receptor expression profile and full list of expressed genes in cholinergic interneurons (ChINs) and spiny projection neurons (SPNs) of the human putamen. Numeric values in counts per million reads.

• Transparent reporting form

### Data availability

RNA sequencing files are available in BioProject with the accession number PRJNA680536.

The following dataset was generated:

| Author(s) | Year | Dataset title | Dataset URL | Database and Identifier |
|---|---|---|---|---|
| Hrabovszky E | 2021 | RNA-seq uncovered detailed transcriptomic profile of cholinergic interneurons and medium spiny projection neurons of the human putamen | http://www.ncbi.nlm.nih.gov/bioproject/680536 | NCBI BioProject, PRJNA680536 |

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
