## [Decision Letter]

**Acceptance summary:**

This multifaceted study reports the discovery of a large population of neurons in the human brain that produce a small peptide molecule known as GnRH. It has been long known that a small population of neurons expressing GnRH in the hypothalamus are central to the onset of puberty and control of adult fertility in both sexes. Here the authors report on a second much larger population of GnRH neurons in a brain region often damaged in neurodegenerative disorders such as Alzheimer’s and Parkinson’s disease. They further report these same neurons express traditional neurotransmitters associated with those diseases, suggesting a novel role for this reproduction-associated peptide.

**Decision letter after peer review:**

Thank you for submitting your article "The cryptic gonadotropin-releasing hormone neuronal system of human basal ganglia" for consideration by *eLife*. Your article has been reviewed by 3 peer reviewers, and the evaluation has been overseen by a Reviewing Editor and Catherine Dulac as the Senior Editor. The following individual involved in review of your submission has agreed to reveal their identity: Tony Plant (Reviewer #3).

Essential revisions:

1) In discussion between the 3 reviewers and reviewing editor it was agreed that the inclusion of data from the mouse model was more detrimental that beneficial and it is recommended it be removed. This is based on the following specific comments: The electrophysiology data in Figure 3 and the Supplementary file are insufficient to assess the quality and outcomes of the studies. No quality parameters such as access resistance or input resistance are provided. Cholinergic neurons are often spontaneously active, so it is difficult to understand why these neurons are reported to be quiescent. Because of the high chloride pipette solution used, all GABA A dependent input that has developed at the age studied will be depolarizing and this adds to the confusion. It is possible the initial membrane potential of these cells is too depolarized for them fire because of inactivation of sodium channels.

To evaluate treatment effects, raw values of membrane potential before and during treatment and washout should be provided in the Supplementary files at minimum and preferably also in figures rather than percent changes. Initial membrane potential needs to be provided on the figures. There are no controls without GnRH application and the initial firing rate in Antide in the example shown is lower so that is difficult to use as a comparator. The lowering of firing rate over time could be due to the very long acclimatization to the whole cell configuration, this could lead currents involved in action potential generation to run down. How data were divided for assessment of firing rate is not indicated. The possibility of dose-dependent changes in the effects of GnRH were not examined and is needed before a purely inhibitory role can be assumed given the GnRH receptor can also couple via Gq, which is typically excitatory.

Perhaps it would be worth the minimal effort to examine whether GnRH neurons in the adult mouse hypothalamus express ChAT. It is stated that such a relationship is not seen in other species but it is unclear whether this is because of negative results or ChAT expression has not been examined.

While I admit that the mouse data are intriguing, direct extrapolation of mouse results to human biology should require more caution. In the human studies, a persistent GnRH population is detected in the striatum of adult individuals, while in the GnRH-GFP mouse, the expression of GFP is transient and actual expression of GnRH is not demonstrated. However, this sets the basis for electrophysiological recordings, whose relevance in terms of human neuronal control is far from clear. Even the fact that Gi components are detected in putative GnRH/ChIN neurons isolated in human brains does not demonstrate the inhibitory nature of the potential GnRH transmission. Admittedly, functional studies in human material are not doable and the use of animal models is a reasonable approach. However, it is not clear to this referee that the mouse data presented here provide a solid back up for the human data, and it is less evident that this should be taken as evidence for an inhibitory action of GnRH in the human striatum, in normal or pathological conditions. Please consider these comments, and edit the presentation and/or discussion of data accordingly.

2) The antisera used were polyclonal and the possibility of overlapping signal produced by a non-decapeptide epitope remains. Because of the striking nature of the claim of over 100-fold more GnRH neurons that previously detected, an additional control of staining with antibodies preabsorbed against the decapeptide form is worth conducting. Related to this, inclusion of data on the expression of THOP1 enzyme that produces this cleavage product in their RNAseq data in addition to the supporting data provided that the biosynthetic enzymes needed to produce the full decapeptide (lines 355-6) or clarification that those lines do include information on the expression THOP1 endopeptidase would facilitate interpretation. A brief statement of what depth color coding accomplishes would be useful to non-expert readers as would discussion of why peptide might not fill processes of these cells as it does for the hypothalamic population. Pre-absorption validation of antibody #1018 should be conducted and presented. Also a low power photomicrograph of the putamen with identifying landmarks and immunostained for GnRH might be included to support the schematic in Figure 1A

3) Initial description of results indicate that anatomical analyses for identification of GnRH neurons was done in three individuals, which are not outright identified. Later, other analyses are applied to other independent samples from other donors. While all the info is detailed in Suppl. Figure 4, which reports a total of 28 individuals in the brain collection, it is suggested the authors make extra-efforts to emphasize how representative the findings are for humans, considering the multiple variables occurring (sex, age, condition, etc.). On the later, were all subjects studied excluded for neurological disease?

4) For the RNAseq, the RIN numbers are somewhat low, and may have contributed to the need for adding an extra cycle for their library creation. The sequencing depth is also adequate but not strong, again likely attributable to the difficulty of this tissue. This should be adequate for a snapshot of presence detection, although one control sample (SPN22) has much lower reads. Notably the expression of GnRH receptor, which is key to their conclusions, is barely above their apparent level of detection at > 5CPM, thus interpretation of these data mandates caution. Inclusion of the sequencing data for the ChIN and SPN marker data would add rigor and also provide the reader with a better idea of the CPM level that can be interpreted. Figure 4 suggests the marker genes are all >60CPM, thelist of "classical receptors" only has 4 (of 77) with average CPM below 5, but the "other receptors" have 67 genes with <5CPM (out of 534) and GNRHR is just >5CPM for the ChIN samples and zero for the SPN. It is also strongly suggested to include the CPM for all the mapped genes in their build (GRCh38.p13).

5) Based on cell numbers and morphology, it is concluded that basal ganglia GnRH neurons have a different origin than hypothalamic GnRH neurons. However, both hypothalamic and extra-hypothalamic GnRH neurons appear to be cholinergic. Did the authors consider performing deep sequencing of hypothalamic GnRH neurons to identify similarities and differences between these two neuronal populations? This may help to better define the differential origin and functional features of these two neuronal sets.

6) The authors openly discuss the fact that the existence of extra-hypothalamic GnRH neuronal populations had been previously defined by pioneering work by Rance and coworkers in 1990's. Admittedly, the present study discloses a much larger population of GnRH neurons, and applies modern technologies to phenotype them. Considering that the previous work identified a much smaller set of extra-hypothalamic GnRH neurons, could the authors speculate why they think this rather large set of GnRH neurons was not identified in that previous study? Was it just because the previous study did not evaluate the striatum? Or was this due to technical limitations of the previous studies?

7) While the present results open numerous questions, the functional implications of the data, in terms of human physiology and pathophysiology, are rather open and the extensive discussion of this issue makes the paper a bit over-speculative. In the same vein, the possibility that GnRH might be considered a useful therapy to neurodegenerative disorders is not sufficiently based on actual data, even considering the mouse results. Application of GnRH analogs is not devoid of potential side-effects and even may suffer desensitization, which makes this possibility quite speculative. I would suggest this sections is markedly shorten if not eliminated.

8) The Discussion is fragmented into multiple sub-sections. While in some cases this is useful to follow the line of reasoning and major conclusions, in other instances, the sub-sections are rather short and mainly recapitulate the Results section. I would suggest that the authors try to integrate the multiple subsections into 2-3 major pieces, so the Discussion appears more fluent and cohesive.

9) Line 72-73 that "GnRH acts on GnRHR1 autoreceptors to regulate higher order non-reproductive functions " is currently only speculation and should be indicated as suc

10) A comment rather than a recommendation – In view of the need for a suitable model for the next generation of studies, the manuscript would benefit from effort into re-examination of extra-hypothalamic neurons in the monkey, rather than pursuing electrophysiology in the mouse.

---

## [Author Response]

Essential revisions:1) In discussion between the 3 reviewers and reviewing editor it was agreed that the inclusion of data from the mouse model was more detrimental that beneficial and it is recommended it be removed. This is based on the following specific comments: The electrophysiology data in Figure 3 and the Supplementary file are insufficient to assess the quality and outcomes of the studies. No quality parameters such as access resistance or input resistance are provided. Cholinergic neurons are often spontaneously active, so it is difficult to understand why these neurons are reported to be quiescent. Because of the high chloride pipette solution used, all GABA A dependent input that has developed at the age studied will be depolarizing and this adds to the confusion. It is possible the initial membrane potential of these cells is too depolarized for them fire because of inactivation of sodium channels.To evaluate treatment effects, raw values of membrane potential before and during treatment and washout should be provided in the Supplementary files at minimum and preferably also in figures rather than percent changes. Initial membrane potential needs to be provided on the figures. There are no controls without GnRH application and the initial firing rate in Antide in the example shown is lower so that is difficult to use as a comparator. The lowering of firing rate over time could be due to the very long acclimatization to the whole cell configuration, this could lead currents involved in action potential generation to run down. How data were divided for assessment of firing rate is not indicated. The possibility of dose-dependent changes in the effects of GnRH were not examined and is needed before a purely inhibitory role can be assumed given the GnRH receptor can also couple via Gq, which is typically excitatory.

We appreciate the critical comments and suggestions about our slice electrophysiology study. We admit that the mechanisms whereby GnRH regulates ChINs in the adult human putamen might be very different from what we have found in the neonatal mouse model. Therefore, we have agreed to leave out the neonatal mouse study (electrophysiology and morphology) from the revised manuscript, as recommended.

Perhaps it would be worth the minimal effort to examine whether GnRH neurons in the adult mouse hypothalamus express ChAT. It is stated that such a relationship is not seen in other species but it is unclear whether this is because of negative results or ChAT expression has not been examined.

As suggested, we have carried out dual-immunofluorescence studies in an attempt to also colocalize ChAT with GnRH in the preoptic area of adult male and female mice. To maximize ChAT signal in this study, we have tried tyramide signal amplification. In spite of technical efforts to increase sensitivity, we have found no evidence for any colocalization between GnRH and ChAT in the mouse preoptic area. To maintain the focus of our revised manuscript (which now has no mouse data) on the human, the negative results of this study on mice are only mentioned briefly in the revised discussion (Lines 352-357) instead of being reported as Result.

While I admit that the mouse data are intriguing, direct extrapolation of mouse results to human biology should require more caution. In the human studies, a persistent GnRH population is detected in the striatum of adult individuals, while in the GnRH-GFP mouse, the expression of GFP is transient and actual expression of GnRH is not demonstrated. However, this sets the basis for electrophysiological recordings, whose relevance in terms of human neuronal control is far from clear. Even the fact that Gi components are detected in putative GnRH/ChIN neurons isolated in human brains does not demonstrate the inhibitory nature of the potential GnRH transmission. Admittedly, functional studies in human material are not doable and the use of animal models is a reasonable approach. However, it is not clear to this referee that the mouse data presented here provide a solid back up for the human data, and it is less evident that this should be taken as evidence for an inhibitory action of GnRH in the human striatum, in normal or pathological conditions. Please consider these comments, and edit the presentation and/or discussion of data accordingly.

We appreciate that the potential value of our electrophysiological work is recognized. Indeed, we continue to believe that our original manuscript provides solid evidence that GnRH at the high 1.2 µM dose transiently inhibits ~50% of ChINs in the CPU of neonatal mice via direct actions on GnRHR1 autoreceptors. However, we accept that these data do not provide sufficient support for the inhibitory role of GnRH in the adult human putamen. All comments regarding inhibition, including the illustration of GNAIs in the revised Figure 3C (RNA-seq results) have been removed from the revised manuscript.

2) The antisera used were polyclonal and the possibility of overlapping signal produced by a non-decapeptide epitope remains. Because of the striking nature of the claim of over 100-fold more GnRH neurons that previously detected, an additional control of staining with antibodies preabsorbed against the decapeptide form is worth conducting.Pre-absorption validation of antibody #1018 should be conducted and presented. Also a low power photomicrograph of the putamen with identifying landmarks and immunostained for GnRH might be included to support the schematic in Figure 1A.

We appreciate the suggestion to carry out the pre-absorption validation of antibody #1018 as further confirmation of labeling specificity. Accordingly, we have verified that pre-absorption of the primary antibody working solution with 0.1 µg/ml GnRH eliminated all labeling from sections of the human putamen. We have described (Lines 89-90; Lines 529-533; Lines 937-938) and illustrated (Figure 2A) these negative results.

In addition, we have prepared and added the requested low-power photomicrograph of the immunostained human basal ganglia with three insets illustrating the position of immunostained GnRH neurons in the putamen (panel B in the revised Figure 1).

Related to this, inclusion of data on the expression of THOP1 enzyme that produces this cleavage product in their RNAseq data in addition to the supporting data provided that the biosynthetic enzymes needed to produce the full decapeptide (lines 355-6) or clarification that those lines do include information on the expression THOP1 endopeptidase would facilitate interpretation.

To our knowledge, only immunohistochemical arguments exist to support the active neurotransmitter role of GnRH1-5 in the extrahypothalamic GnRH neurons in the developing monkey brain. These neurons cannot be immunostained with several GnRH antibodies, including #LR-1. We have shown that this argument is not relevant to the human putamen by showing that GAP and several GnRH antibodies, including #LR-1, do recognize well human extrahypothalamic GnRH neurons. An additional argument used in the previous studies on embryonic/fetal monkeys by the Terasawa group was the immunohistochemical detection of THOP1 in extrahypothalamic GnRH neurons. In the revised manuscript we have discussed more abundantly the ubiquitous nature and multiple substrates of THOP1. We argued that presence of THOP1 in human ChINs does not necessarily indicate that GnRH is cleaved and inactivated by this enzyme. We supported this view by noting that Terasawa and co-workers also identified THOP1 signal in the developing *bona fide* hypothalamic GnRH neurons (late type GnRH neurons) of the monkey which can undoubtedly generate and release the GnRH decapeptide. We agree with the reviewer that our data do not exclude the possibility that GnRH1-5 also has a functional role. This has also been recognized in the revised and extended discussion. (Lines 269-303).

A brief statement of what depth color coding accomplishes would be useful to non-expert readers as would discussion of why peptide might not fill processes of these cells as it does for the hypothalamic population.

We have added a brief explanatory statement about depth color coding, as suggested (Line 920-921).

We have offered two possible explanations for the insufficient dendritic labeling of ChINs (Lines 145-146). (“This limitation could be due to the low amount and/or restricted subcellular distribution of the peptide.”)

3) Initial description of results indicate that anatomical analyses for identification of GnRH neurons was done in three individuals, which are not outright identified. Later, other analyses are applied to other independent samples from other donors. While all the info is detailed in Suppl. Figure 4, which reports a total of 28 individuals in the brain collection, it is suggested the authors make extra-efforts to emphasize how representative the findings are for humans, considering the multiple variables occurring (sex, age, condition, etc.). On the later, were all subjects studied excluded for neurological disease?

We agree that a large biological heterogeneity characterizes the human samples which is complicated further by technical variations that are difficult to control (e.g. *perimortem* and *postmortem* conditions). Such conditions may all influence the staining and quantification results and, unfortunately, it is not possible to determine the contribution of individual variables to the staining results obtained in individual samples. In the revised manuscript we have recognized the existence of sample heterogeneity (Lines 308-311; Lines 447-449). We have also stated that samples were obtained from individuals without known neurological disorders (Line 445). For each study, we have now specified in Results the code and number of samples used for each experiment. The revised Supplementary File 1 includes data on the human subjects used in each experiment.

4) For the RNAseq, the RIN numbers are somewhat low, and may have contributed to the need for adding an extra cycle for their library creation. The sequencing depth is also adequate but not strong, again likely attributable to the difficulty of this tissue. This should be adequate for a snapshot of presence detection, although one control sample (SPN22) has much lower reads. Notably the expression of GnRH receptor, which is key to their conclusions, is barely above their apparent level of detection at > 5CPM, thus interpretation of these data mandates caution. Inclusion of the sequencing data for the ChIN and SPN marker data would add rigor and also provide the reader with a better idea of the CPM level that can be interpreted. Figure 4 suggests the marker genes are all >60CPM, thelist of "classical receptors" only has 4 (of 77) with average CPM below 5, but the "other receptors" have 67 genes with <5CPM (out of 534) and GNRHR is just >5CPM for the ChIN samples and zero for the SPN. It is also strongly suggested to include the CPM for all the mapped genes in their build (GRCh38.p13).

We appreciate the technical comments about RNA-sequencing. We agree that, in general, suboptimal methods may cause lack of detection of low-abundance transcripts. In the past few years our laboratory invested much effort into the development of an optimal protocol to carry out RNA-seq studies on cell populations isolated with LCM from formalin-fixed *postmortem* samples. The selection of the “FFPE” RNA isolation kit and the RNA-seq library preparation kit (which latter uses ribosomal RNA depletion and random primers instead of oligo(dT), for reverse transcription) allow reliable library preparation from nanogram amounts of partly degraded starting RNA (Schuierer et al., 2017). We believe that the 13664 and 12637 different transcripts we identified at CPM≥5 in ChINs and the 13,558 and 13,682 transcripts at CPM≥5 in SPNs, indicate a reasonably high detection sensitivity of the method for even low abundance receptor transcripts, including *GNRHR1*. *GNRHR1* occurred in both independent human ChIN samples at similar relative abundances (i.e. cpm 7.8 and 5.0), while being entirely absent from the two SPN pools.

We have agreed to provide a complete RNA-seq dataset. We have supplemented the revised Supplementary File 3 which now has a sheet with the full list of annotated ChIN and SPN transcripts, irrespective of cpm.

5) Based on cell numbers and morphology, it is concluded that basal ganglia GnRH neurons have a different origin than hypothalamic GnRH neurons. However, both hypothalamic and extra-hypothalamic GnRH neurons appear to be cholinergic. Did the authors consider performing deep sequencing of hypothalamic GnRH neurons to identify similarities and differences between these two neuronal populations? This may help to better define the differential origin and functional features of these two neuronal sets.

We appreciate this suggestion and we agree that such a comparison would be critically important and interesting. A promising method for deep transcriptome profiling of immunohistochemically labeled hypothalamic GnRH neurons is currently under development in our laboratory. A major technical challenge is still to preserve RNA during immunohistochemical procedures. We have recognized this challenge in the revised Discussion (Lines 333-340).

6) The authors openly discuss the fact that the existence of extra-hypothalamic GnRH neuronal populations had been previously defined by pioneering work by Rance and coworkers in 1990's. Admittedly, the present study discloses a much larger population of GnRH neurons, and applies modern technologies to phenotype them. Considering that the previous work identified a much smaller set of extra-hypothalamic GnRH neurons, could the authors speculate why they think this rather large set of GnRH neurons was not identified in that previous study? Was it just because the previous study did not evaluate the striatum? Or was this due to technical limitations of the previous studies?

In the revised manuscript, we have discussed that Rance of coworkers did not analyze the entire extent of the putamen. We believe that, in itself, this may explain why the cell numbers differ so much in the two studies (Lines 311-315). In addition, we have discussed why we think that our immunohistochemical study could even fail to detect neurons with low signal levels; therefore, we suggest that the real number of extrahypothalamic GnRH neurons could be even higher than in our report (Lines 308-311).

7) While the present results open numerous questions, the functional implications of the data, in terms of human physiology and pathophysiology, are rather open and the extensive discussion of this issue makes the paper a bit over-speculative. In the same vein, the possibility that GnRH might be considered a useful therapy to neurodegenerative disorders is not sufficiently based on actual data, even considering the mouse results. Application of GnRH analogs is not devoid of potential side-effects and even may suffer desensitization, which makes this possibility quite speculative. I would suggest this sections is markedly shorten if not eliminated.

As suggested, we have omitted the speculation about the clinical use of GnRH analogs from the abstract and the discussion. We note that although the reproductive side effects of GnRH analogues would limit the use of this strategy in clinical practice (Almeida, 2004), the transcriptome profile of ChINs (Figure 5 and Supplementary File 3) offers a few alternative receptorial mechanisms to counteract the hyperactivity of ChINs in PD (Lines 424-427).

8) The Discussion is fragmented into multiple sub-sections. While in some cases this is useful to follow the line of reasoning and major conclusions, in other instances, the sub-sections are rather short and mainly recapitulate the Results section. I would suggest that the authors try to integrate the multiple subsections into 2-3 major pieces, so the Discussion appears more fluent and cohesive.

As recommended, we combined several subsections where we could in the revised Discussion to achieve an improved flow.

9) Line 72-73 that "GnRH acts on GnRHR1 autoreceptors to regulate higher order non-reproductive functions " is currently only speculation and should be indicated as suc

By removing the whole mouse block, we have given up the only functional evidence for the direct action of GnRH on GnRHR1 autoreceptors in ChINs. The idea that GnRH acts on GnRHR1 autoreceptors now only relies on the RNA-seq detection of both *GNRH1* and *GNRHR1* in human ChINs but not in SPNs. Although we are confident that the RNA-seq data are strong, we use conservative writing when suggesting that GnRH likely acts on autoreceptors in the human ChINs. In addition, we also recognize the possibility that other rare cell types of the human putamen not studied here may also express GnRHR1 (Lines 389-391).

10) A comment rather than a recommendation – In view of the need for a suitable model for the next generation of studies, the manuscript would benefit from effort into re-examination of extra-hypothalamic neurons in the monkey, rather than pursuing electrophysiology in the mouse.

We also appreciated this comment and we agree that monkeys would provide a much better model animal than neonatal mice. Using HPLC-MS/MS, it would be interesting and relatively easy to revisit the issue of whether extrahypothalamic brain regions of the monkey have GnRH, GnRH1-5, or both.